# Prediction error determines how memories are organized in the brain

Nicholas GW Kennedy[1], Jessica C Lee[1,2], Simon Killcross[1], R Fred Westbrook[1], Nathan M Holmes[1]*

[1]School of Psychology, University of New South Wales, Sydney, Australia; [2]School of Psychology, University of Sydney, Sydney, Australia

**Abstract** How is new information organized in memory? According to latent state theories, this is determined by the level of surprise, or prediction error, generated by the new information: a small prediction error leads to the updating of existing memory, large prediction error leads to encoding of a new memory. We tested this idea using a protocol in which rats were first conditioned to fear a stimulus paired with shock. The stimulus was then gradually extinguished by progressively reducing the shock intensity until the stimulus was presented alone. Consistent with latent state theories, this gradual extinction protocol (small prediction errors) was better than standard extinction (large prediction errors) in producing long-term suppression of fear responses, and the benefit of gradual extinction was due to updating of the conditioning memory with information about extinction. Thus, prediction error determines how new information is organized in memory, and latent state theories adequately describe the ways in which this occurs.

## eLife assessment

This is a **fundamental** study examining the role of prediction error in state allocation of memories. The data provided are **convincing** and largely support the conclusion that a gradual change between acquisition and extinction maintains the memory state of acquisition and thus results in extinction that is resistant to restoration. This paper is of interest to behavioural and neuroscience researchers studying learning, memory, and the neural mechanisms of those processes as well as to clinicians using extinction-based therapies in treating anxiety-based disorders

*For correspondence:
n.holmes@unsw.edu.au

## Introduction

Learning about the relations between events in the environment is a fundamental aspect of animal and human life. It allows organisms to use the past to predict the future and, thereby, behave appropriately in the present. For example, upon hearing a distinctive sound that had previously signaled the presence of a predator, organisms engage in defensive responses that serve to avoid the imminent danger; and upon detecting a distinctive smell that had previously signaled the presence of food, organisms engage in approach responses that procure the food. However, environments change, and events that previously served as reliable signals for biologically significant states-of-affairs may cease to do so: the sound may not be followed by the arrival of the predator and the smell might be present but the food absent. How is this new information organized with respect to the existing knowledge? One possibility is that it is directly incorporated into the original knowledge, resulting in some form of overwriting (the sound and the smell are now encoded as less reliable signals for their associated consequences). Alternatively, the new information may be classified as distinct from the original knowledge, resulting in separate storage of the new information (the sound and smell predict

**Figure 1.** *Cochran and Cisler, 2019* latent state model simulation of standard extinction with spontaneous recovery. The simulations depict the associative strength (**A**) and latent state beliefs (**B**) of a conditioned stimulus (CS) across conditioning, extinction, and spontaneous recovery. Initially, there are 10 conditioning trials (t=1–10, R=1), followed by 80 extinction trials (t=11–90, R=0). A time delay was inserted between trials 90 and 91 to simulate spontaneous recovery before extinction trials continued (t=91–110, R=0). The CS's associative strength (**A**) increases across conditioning trials, sharply decreases across extinction trials, and briefly recovers post time delay, indicating a spontaneous recovery of performance. The right panel (**B**) depicts the degree of belief that the most recent latent state is active. If the belief changes throughout the simulation, this indicates that a new latent state has been inferred. Upon the beginning of extinction (t=11), the figure indicates that a new state has been inferred (due to the large prediction error generated by the absence of the unconditioned stimulus, US). The simulation captures the decrement of performance associated with extinction as well as the recovery of performance that occurs after a time delay.

danger and food in some circumstances, but not others). However, it is not currently clear what factors determine the use of either processing strategy.

Many contemporary models of learning suggest that prediction error influences how new information is organized in memory. These models (*Cochran and Cisler, 2019*; *Gershman et al., 2010*; *Gershman et al., 2017*; *Redish et al., 2007*) hold that animals and people encode information about events/contexts and assign this information to a particular 'state' of the memory system. Critically, this assignment of information is regulated by prediction error: that is, the difference between what is predicted to happen based on one's existing knowledge and what actually happens. These so-called latent state models are like classic associative models (e.g. *Pearce and Hall, 1980*; *Rescorla and Wagner, 1972*) in holding that prediction error drives learning, such that the greater the prediction error, the greater the amount of associative change. However, they differentiate themselves from earlier models in proposing that prediction error is also an index of similarity to past experiences, which then influences the assignment of event and context information to memory. When a new experience is similar to a past experience and, hence, the prediction error is small, the new experience is integrated into the same memory state as the past experience, resulting in updating (or overwriting) of existing information. By contrast, when a new experience is different to past experiences and, hence, the prediction error is large, the new experience is encoded into a new state, resulting in the preservation of existing information.

Extinction of Pavlovian conditioned fear in rats is a laboratory protocol that has been used to study how new information is integrated with existing knowledge (e.g. *Dunsmoor et al., 2015*; *Gershman et al., 2013*). In this protocol, rats first learn that a conditioned stimulus (CS [e.g. a tone]) signals a brief-but-aversive unconditioned stimulus (US [e.g. a foot-shock]). They are then exposed to repeated presentations of the CS alone, across which conditioned fear responses (e.g. potentiated startle, freezing) decline in frequency or vigor until they cease altogether, at which point they are said to be extinguished. Importantly, this extinction of conditioned fear is not due to unlearning of the CS-US association, as evidenced by findings that fear responses can be restored through a range of post-extinction manipulations including the passage of time (spontaneous recovery), re-exposure to the

US (reinstatement), or exposure to the CS outside of its extinction context (renewal). Instead, such findings have been taken to imply that extinction involves new learning (e.g. CS-no US) that leaves the original conditioning intact but interferes with its expression in behavior (*Bouton, 1993*; *Bouton, 2004*).

According to latent state models, the CS-US association survives extinction (and thereby permits fear restoration) because of so-called 'state-splitting' triggered by the abrupt removal of the US. One such model is that proposed by *Cochran and Cisler, 2019*. This model not only explains fear restoration phenomena but also a wider range of conditioning phenomena than other latent state models (e.g. *Redish et al., 2007*; *Gershman et al., 2010*: see *Cochran and Cisler, 2019*, for comparisons). Consequently, we use the Cochran-Cisler model here as an exemplar. According to this model, the shift from conditioning to extinction causes the subject to experience a large prediction error: hence, they infer that a new state is active (this is the state-split) and encode the extinction experience into this state, separate from that of conditioning. Importantly, the subject is also biased towards assuming that the most recent latent state is active; however, this bias decreases with time outside of the experimental context and, with sufficient time, all states return to equal probability of activation. Thus, during extinction, the subject continues to believe the extinction state is active, however, when the subject is re-exposed to the extinguished CS after a significant passage of time, the conditioning state has a greater likelihood of activation (relative to the same likelihood at the end of extinction), resulting in recovery – that is, spontaneous recovery – of conditioned responding (for simulations, see *Figure 1A and B*).

An implication of the *Cochran and Cisler, 2019* model is that, if the prediction error on the initial trials of extinction is small, the CS-alone experiences would be assigned to the same state as that of the prior CS-US experiences and more effectively impair recall of conditioning. In practice, small prediction errors could be achieved by progressively and gradually reducing the intensity of the US across sessions prior to presentations of the CS alone (e.g. shifting from 0.8 mA to 0.4 mA, to 0.2 mA, to 0.1 mA, to no US). If the series of small prediction errors is sufficient to produce learning but prevent state-splitting, then manipulations that typically recall the CS-US experiences should be less effective in doing so and, thus, evoke less responding. Thus, in contrast to the standard case where CS-US and CS alone experiences are encoded in different states, the gradual case permits CS-US and CS alone experiences to be encoded in the same state. Hence, the gradual case should more effectively undermine recovery and/or reinstatement of responding to the CS (for simulations, see *Figure 2B and C*).

The present experiments tested this implication of the *Cochran and Cisler, 2019* model. Specifically, they examined whether a gradual extinction protocol (one which generates small prediction errors) produces a combined conditioning and extinction state and, thereby, less spontaneous recovery and reinstatement of responding with the passage of time or US re-exposure, respectively. The initial experiments confirmed that gradual extinction does, indeed, have these effects, as predicted by the Cochran-Cisler model and other latent state models (e.g. *Gershman et al., 2010*). Subsequent experiments examined the conditions under which this gradual extinction effect is observed. Finally, we attempted to explain the full pattern of results in these experiments using simulations of the Cochran-Cisler model.

## Results
### Experiment 1: Gradual extinction produces less spontaneous recovery of fear than standard extinction

Experiment 1 assessed the prediction of the *Cochran and Cisler, 2019* latent state model that gradual extinction results in less spontaneous recovery than standard extinction. Simulations (*Figure 2B and C*) confirmed this prediction. To assess this prediction, all rats received fear conditioning, consisting of four pairings of a tone CS and foot shock US (0.8 mA × 0.5 s). They were then allocated randomly to two extinction conditions. Rats in the first condition (standard extinction) received CS alone presentations each day across three days, while those in the second condition (gradual extinction) continued to receive pairings of the CS and shock US across these days. The duration of the shock remained at 0.5 s, but, critically, the intensity of the shock was reduced from one day to the next: from 0.4 mA on day one, to 0.2 mA on day two and, finally to 0.1 mA on day three. Rats in both conditions then

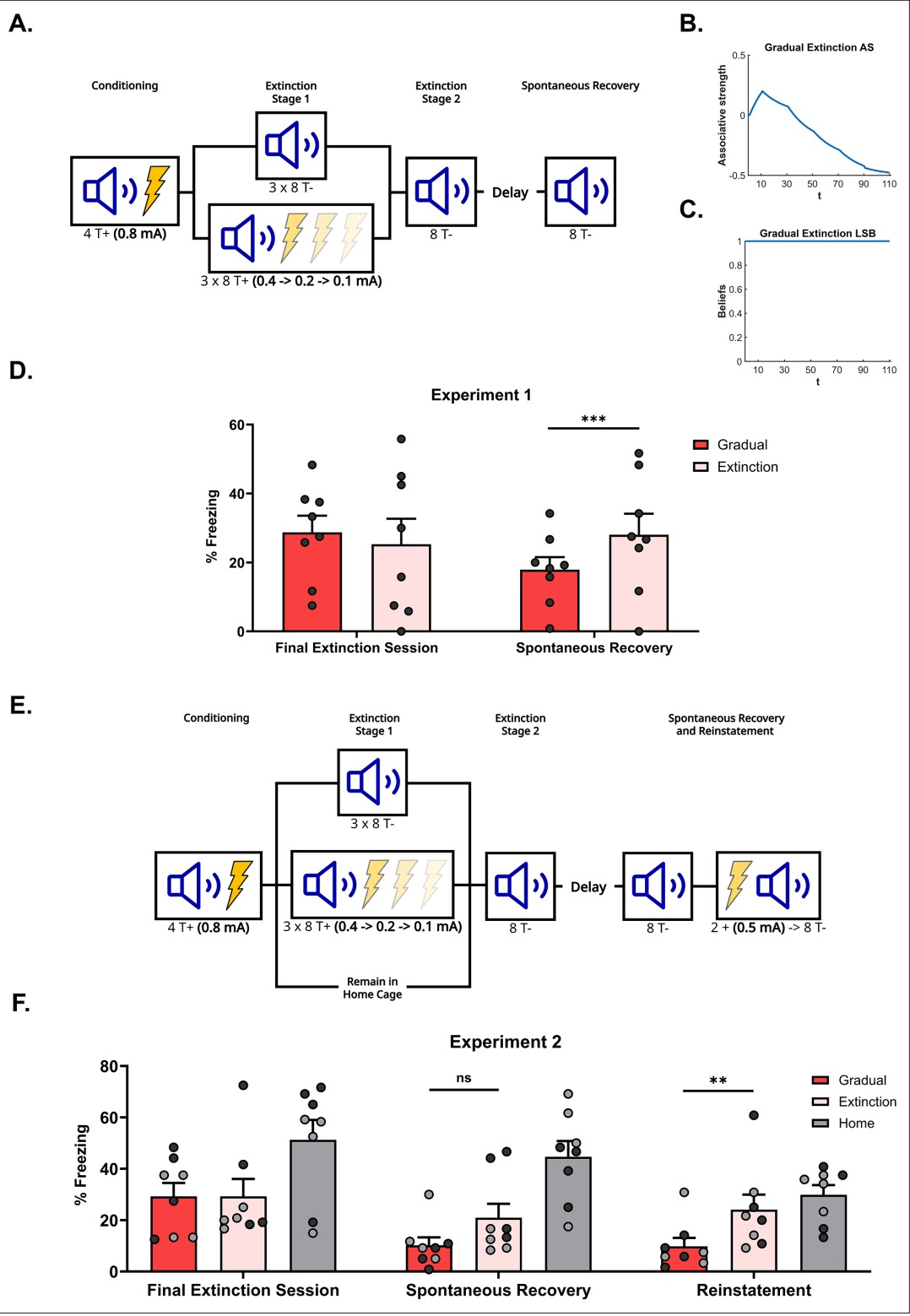

**Figure 2.** Gradual extinction is more effective than standard extinction. (**A**) Design for Experiment 1: Rats were conditioned with tone-shock pairings and then either received standard extinction (Group Extinction, n=8, tone alone presentations) or gradual extinction (Group Gradual, n=8, tone-shock pairings – shock intensity decreased across the day). Following, all groups received standard extinction of the tone and were tested for spontaneous recovery two weeks later. Simulation of the latent state model showing the associative strength (**B**) and latent state beliefs (**C**) of a gradually extinguished conditioned stimulus (CS) across conditioning, extinction, and

*Figure 2 continued on next page*

*Figure 2 continued*

spontaneous recovery (*Cochran and Cisler, 2019*). Initially, there are 10 conditioning trials (t=1–10, *R*=1), followed by 80 extinction trials (t=11–90). Within the (gradual) extinction trials, the CS is still paired with the unconditioned stimulus (US), however, the intensity of the US is halved every 20 trials (t=11–30, *R*=0.5; t=31–50, *R*=0.25; t=51–70, *R*=0.125) until it is removed for the final 20 extinction trials (t=71–90, *R*=0). A time delay was inserted between trials 90 and 91, simulating spontaneous recovery, before a final 20 extinction trials (t=91–110, *R*=0). Associative strength (**B**) increases across conditioning, decreases steadily across gradual extinction, and remains low after a time delay, indicating an absence of spontaneous recovery. Panel (**C**) depicts the degree of belief that the most recent latent state is active. If the belief changes throughout the simulation, this indicates that a new latent state has been inferred. The belief remains at 1 across conditioning, extinction, and spontaneous recovery, indicating that the prediction errors produced by gradual extinction are not sufficient to produce state-splitting. Full simulation details are included in Appendix 3. (**D**) Percentage freezing levels across the final session of extinction (left) and spontaneous recovery (right). Bars represent means ± SEM. Dots represent individual freezing levels (males = dark gray, females = light gray). Freezing levels were similar between groups at the final session of extinction, however, Group Gradual displayed lower freezing than Group Extinction at the spontaneous recovery test. (**E**) Design for Experiment 2: Rats were conditioned with tone-shock pairings and then either received standard extinction (Group Extinction, n=8), gradual extinction (Group Gradual, n=8), or remained in their home cage (Group Home, n=8). Following, all groups received standard extinction of the tone and were tested for spontaneous recovery and reinstatement. (**F**) Percentage freezing levels across the final session of extinction (left), spontaneous recovery test (middle), and reinstatement test (right). Bars represent means ± SEM. Dots represent individual freezing levels (males = dark gray, females = light gray). Freezing levels were similar between Groups Gradual and Extinction at the final session of extinction, while Group Home displayed a higher level. Groups Gradual and Extinction displayed less freezing than Group Home at both the spontaneous recovery and reinstatement test. However, while Group Gradual showed less freezing than Group Extinction at both tests, the difference only reached significance at the reinstatement test.

received CS alone presentations across two further days. Finally, two weeks later, rats in both conditions were tested for spontaneous recovery of fear responses (freezing) to the extinguished CS (see *Figure 2A* for design).

Here and in subsequent experiments, results are reported for the data of interest; specifically, the levels of freezing elicited by the CS in the final session of extinction and the tests for spontaneous recovery and/or reinstatement. Figures and statistics for freezing levels across conditioning and extinction are included in Appendix 1.

*Figure 2D* shows mean freezing (±standard error of the mean [SEM]) across the final session of extinction and the spontaneous recovery test. During the final session of extinction, rats in both conditions displayed a moderate level of freezing, and there was no significant difference between the two groups ($F < 1$). By contrast, in the spontaneous recovery test, rats in Group Gradual froze significantly less than those in Group Extinction ($F_{1,13} = 5.040$, $p = 0.043$, $d = 1.158$, 95% CI [0.044, 2.280]), suggesting that gradual extinction had produced more robust extinction learning than standard extinction. This accords with the simulations of the *Cochran and Cisler, 2019* latent state model which holds that rats which received gradual extinction encoded their conditioning and extinction experiences into a single state, while rats which received standard extinction encoded their conditioning and extinction into separate states.

## Experiment 2: Gradual extinction produces less reinstatement of fear than standard extinction

Experiment 2 had two aims. The first was to replicate the results of the previous experiment by again using the spontaneous recovery test to compare the long-term effectiveness of gradual extinction relative to standard extinction. We additionally included a control group to provide some indication of the level of spontaneously recovered responding among rats exposed to gradual or standard extinction. The second aim was to assess the generality of the results from Experiment 1 by using a reinstatement manipulation to restore responding to the extinguished tone CS. Reinstatement typically consists of re-exposing rats to the US (shock) prior to testing of the CS alone. According to latent state models, the occurrence of the US after standard extinction is taken to imply that the conditioning state is again active, hence conditioned responses return. The hypothesis to be tested is that, among rats trained in the gradual extinction protocol, the occurrence of the shock is less likely to reinstate

responding to the extinguished tone because the conditioning and extinction experiences occupy the same state: hence, they cannot be easily disambiguated.

The design is shown in *Figure 2E*. Three groups of rats received pairings of the tone CS and a 0.8 mA × 0.5 s foot shock US followed by extinction. For rats in Group Gradual, the intensity of the shock US paired with the CS was progressively reduced across the first three days of the extinction stage, and this was followed by a fourth day on which the CS was presented alone. For rats in Group Extinction, the extinction stage involved four days of CS alone exposure. Finally, rats in Group Home remained in their home cage across the first three days of this stage, before receiving a session of CS alone presentations on the fourth day. Two weeks later, all rats were tested for spontaneous recovery of extinguished responses, and the following day, for their reinstatement after a 0.5 mA × 0.5 s shock alone presentation.

*Figure 2F* shows mean freezing (±SEM) elicited by the CS across the final session of extinction (left), the spontaneous recovery test (middle), and the reinstatement test (right). In the final session of extinction, rats in Group Home froze significantly more to the CS than those in Groups Gradual and Extinction ($F_{1,21} = 7.237$, $p = 0.014$, $d = 1.171$, $95\% CI [-2.065, -0.264]$). There was no significant difference in freezing between the latter two groups ($F < 0.1$) indicating that the gradual and standard extinction treatments were ultimately, equally effective in reducing conditioned fear responses.

In the test for spontaneous recovery two weeks later, rats in Group Home again froze significantly more to the CS than those in Groups Extinction and Gradual ($F_{1,21} = 22.113$, $p < 0.001$, $d = 1.886$, $95\% CI [-2.937, -1.136]$), indicating that fear responses only partially recovered in the latter groups. Despite the numerical difference, Group's Gradual and Extinction did not significantly differ from each other in this test ($F < 3$). The lack of statistical significance is likely due to low levels of spontaneous recovery in these groups, masking any differences between them. This is supported by the finding that both Group Extinction and Group Gradual displayed a decrease in freezing from the final extinction session to the spontaneous recovery test, perhaps due to the treatments having failed to completely extinguish responding.

In the reinstatement test, CS freezing was significantly higher in Group Home compared to Groups Extinction and Gradual ($F_{1,21} = 5.702$, $p = 0.026$, $d = 1.003$, $95\% CI [-1.934, -0.133]$) and, importantly, in Group Extinction compared to Group Gradual ($F_{1,21} = 5.178$, $p = 0.033$, $d = 0.63$, $95\% CI [0.098, 2.178]$). Thus, the reinstatement test revealed more effective extinction learning in Groups Gradual and Extinction than in Group Home and, importantly, more effective learning in Group Gradual than Group extinction.

## Experiment 3: The effectiveness of gradual extinction depends on a progressive reduction in shock intensity, not on a reduction in shock intensity

Experiment 1 demonstrated that a gradual extinction protocol produced less spontaneous recovery than standard extinction and Experiment 2 showed the same pattern for reinstatement. According to the *Cochran and Cisler, 2019* model, the effectiveness of gradual extinction is due to the series of small prediction errors, created via the progressive reduction in US intensity from that used to condition the CS to the final CS alone presentations. However, the designs of the previous experiments confounded the progressive reductions in shock intensity with exposure to lower-intensity shocks. It is possible that simply receiving lower-intensity shocks following conditioning, regardless of their order, is sufficient to produce the pattern of results observed thus far. If this were to be the case, then presenting the same number and intensity of CS-shock pairings, but in a scrambled order, would produce the same robustness of extinction as the progressive reduction in shock intensities. However, the Cochran-Cisler model predicts that the prediction errors produced by the scrambled CS-shock pairings would, like standard extinction, produce state-splitting, and thus, would not produce the same robustness of extinction. Simulations confirmed this prediction (*Figure 3B and C*).

Experiment 3 tested this prediction (see *Figure 3A* for design). There were three groups. The gradual and standard extinction groups differed from the previous experiments only in that the initial three days of the extinction stage was followed by six daily sessions of CS alone presentations (as opposed to the two and one days of CS alone presentations used in Experiments 1 and 2). The six additional days of extinction were included to ensure that conditioned freezing was more fully extinguished than in the previous experiments. The third group (Scrambled) continued to receive CS-shock

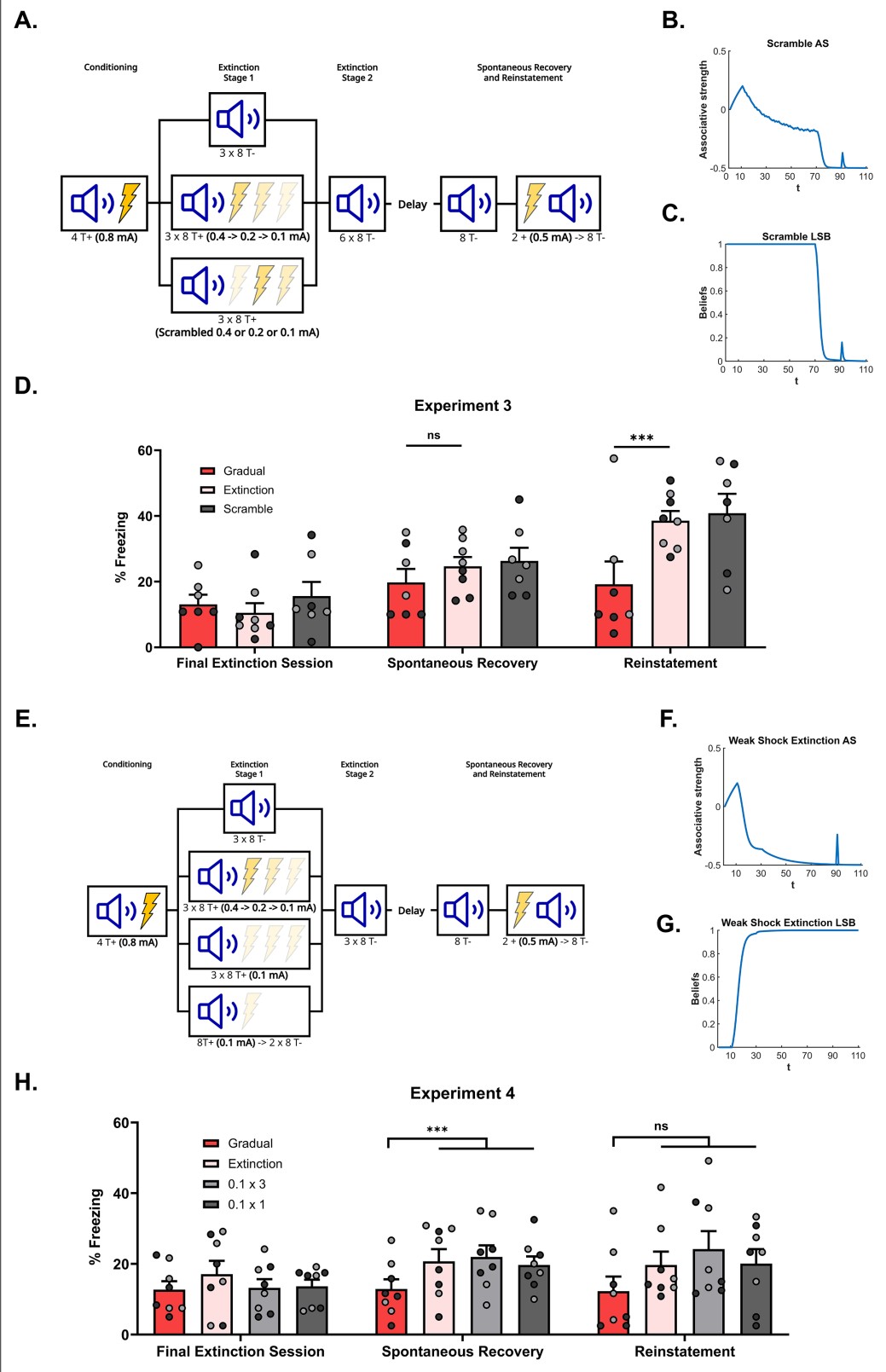

**Figure 3.** The effectiveness of gradual extinction is dependent on a progressive reduction in the shock intensity.
(**A**) Design for Experiment 3: Rats were conditioned with tone-shock pairings and then either received standard extinction (Group Extinction, n=8), gradual extinction (Group Gradual, n=8), or scrambled extinction (Group Scramble, n=8 – same number and intensity of tone-shock pairings as Group Gradual but in a pseudo-random

*Figure 3 continued on next page*

*Figure 3 continued*

order). Following, all groups received standard extinction of the tone and were tested for spontaneous recovery and reinstatement two weeks later. Simulation of the latent state model showing the associative strength (**B**) and latent state beliefs (**C**) of Group Scramble across conditioning, extinction, and spontaneous recovery (*Cochran and Cisler, 2019*). The trial structure remained identical to the structure used in the simulation displayed in *Figure 2C and D* with the exception of the 60 (gradual) extinction trials. These trials were the same in number and intensity as Group Gradual (i.e. 20 trials each of $R=0.5$, $R=0.25$, and $R=0.125$) but were now arranged in a pseudo-random order. The order was identical to the order used in the corresponding experiment (see Methods experiment 3 for exact sequence). The standard extinction and spontaneous recovery trials remained the same. Associative strength increases across conditioning, declines across extinction at a slow rate, with a lot of trial-to-trial variability. It declines further across the final extinction trials before recovering at the spontaneous recovery test. Latent state beliefs (**C**) switched upon the removal of the reward ($t=71$, $R=0$) indicating that a new latent state had been inferred. Thus, the simulations show that scrambled extinction is not as effective as gradual extinction in producing robust extinction learning. (**D**) Percentage freezing levels across the final session of extinction (left), spontaneous recovery test (middle), and reinstatement test (right). Bars represent means ± SEM. Dots represent individual freezing levels (males = dark gray, females = light gray). Freezing levels are similar across the final session of extinction and the spontaneous recovery test for all groups. Group Gradual displayed less freezing than Groups Extinction and Scramble at the reinstatement test. (**E**) Design for Experiment 4: Rats were conditioned with tone-shock pairings and then either received standard extinction (Group Extinction, n=8), gradual extinction (Group Gradual, n=8) or weak shock extinction. Those who received weak shock extinction were split into two groups who received either 1 or 3 days of pairings (Group 0.1 × 3, n=8 – Three days of tone-shock pairings at the weakest shock intensity; Group 0.1 × 1, n=8 – A single day of tone-shock pairings at the weakest shock intensity and two days of tone-alone presentations). Following, all groups received standard extinction of the tone and were tested for spontaneous recovery and reinstatement two weeks later. Simulation of the latent state model showing the associative strength (**F**) and latent state beliefs (**G**) of Group 0.1 × 3 across conditioning, extinction, and spontaneous recovery (*Cochran and Cisler, 2019*). The trial structure remained identical to the structure used in the simulation displayed in *Figure 2C and D* with the exception of the 60 gradual extinction trials. All gradual extinction trials now immediately shifted to the lowest intensity reward ($t=11$–70, $R=.125$). The remaining extinction and spontaneous recovery trials remained the same. Associative strength (**F**) increases across conditioning, decreases quickly across the weak shock extinction trials and standard extinction trials before recovering at the spontaneous recovery test. Latent state beliefs (**G**) change upon the first trial of weak shock extinction, indicating that a new latent state had been inferred. Thus, the simulations show that weak shock extinction is not as effective as gradual extinction in producing robust extinction learning. (**H**) Percentage freezing levels across the final session of extinction (left), spontaneous recovery test (middle), and reinstatement test (right). Bars represent means ± SEM. Dots represent individual freezing levels (males = dark gray, females = light gray). Freezing levels are similar across the final session of extinction for all groups. Group Gradual displayed less freezing than all other groups at the spontaneous recovery and reinstatement tests, but this difference only reached statistical significance at the spontaneous recovery test.

---

pairings across the initial three days of the extinction stage. The number of pairings and intensities of the shocks were identical to those received by the gradual group but the order in which the intensity of the shock changed was scrambled rather than progressively reducing as in the gradual group. Specifically, trials on which the CS was paired with the 0.4, 0.2, or 0.1 mA shock were delivered in a pseudo-random order within each session, (e.g. a 0.4 mA pairing followed by a 0.1 mA and then a 0.2 mA, etc.). This scrambled group then received six sessions of CS alone presentations as described for the other two groups. Two weeks later, all groups were tested for spontaneous recovery of freezing to the CS, and this was followed by a final test for reinstatement of freezing following shock re-exposure.

*Figure 3D* shows mean freezing (±SEM) across the final session of extinction (left), the spontaneous recovery test (middle), and the reinstatement test (right). In the final session of extinction and spontaneous recovery test, the level of freezing did not differ between the three groups ($Fs < 2$). Once again, the overall level of spontaneous recovery was very low which may have masked differences between the groups; the daily CS alone presentations across the additional six days may have over-extinguished the CS. By contrast, in the reinstatement test, the level of freezing to the CS was significantly lower in Group Gradual compared to Groups Extinction and Scrambled ($F_{1,19} = 9.598$, $p = 0.006$, $d = 1.326$, $95\% \, CI \, [0.460, 2.378]$). Freezing did not significantly differ in the latter two groups ($Fs < 2$). Thus, despite the lack of difference in the spontaneous recovery test, Group Gradual exhibited less reinstatement than Groups Extinction and Scrambled, replicating the result of Experiment 2 and, importantly, showing that exposure to the lower-intensity shocks in a

pseudo-random or scrambled order is not sufficient to decrease fear restoration. Rather, the series of small prediction errors, created by progressively reducing shock intensities, is critical.

## Experiment 4: The effectiveness of gradual extinction is not due to shock devaluation

The previous experiments leave open the possibility that the gradual extinction effect is due to the devaluation of the US representation rather than any change in prediction error. That is, after conditioning, the subject presumably holds a representation of the CS and the shock US. Receiving the series of lower-intensity shocks following conditioning degrades the subject's shock representation, such that the shock comes to be represented as very weak. Thus, recall of the CS-shock association in tests for spontaneous recovery (Experiment 1) or reinstatement (Experiments 2 and 3) elicits little fear. This explanation differs from that of the *Cochran and Cisler, 2019* model in that degradation of the shock representation is not bound to the size of any prediction error. Rather, it implies that shifting immediately to a very low-intensity shock (0.1 mA) after conditioning with a high-intensity shock (0.8 mA) would be sufficient to protect against spontaneous recovery and/or reinstatement. By contrast, the Cochran-Cisler model predicts that shifting immediately to a very low-intensity shock (0.1 mA) would still produce a large prediction error, resulting in state-splitting and, thereby, susceptibility to fear restoration. Simulations in relation to this prediction are shown in *Figure 3F and G*.

Experiment 4 tested this prediction (see *Figure 3E* for design). There were four groups. The gradual and standard extinction groups differed from those in the previous experiments only in that the initial three days of the extinction stage was followed by three daily sessions of CS alone presentations. Three days were chosen as an intermediary between one and six, as we assumed that those amounts of extinction under-extinguished or over-extinguished the CS, respectively. The treatment of the remaining two groups differed from that of Groups Gradual and Extinction only in the initial three days of the extinction stage. One group (0.1 × 3) received pairings of the CS and the lowest intensity shock, 0.1 mA, on each of these days, matching the total amount of shocks administered to Group Gradual. The final group (0.1 × 1) received pairings of the CS and a 0.1 mA shock for only the first of these days, and CS alone presentations for the remaining two days. This group was included to test whether a single session of low-intensity shocks was sufficient to protect against fear restoration. After the final three days of the extinction stage, across which all groups received CS alone presentations, all groups were tested for spontaneous recovery of freezing to the CS. The interval to the spontaneous recovery test was increased to three weeks (from two weeks in the previous experiments) in an attempt to increase recovery of the extinguished CS. The following day, all groups received a shock-alone presentation and a test for reinstatement of extinguished freezing.

*Figure 3H* shows mean freezing (±SEM) across the final session of extinction (left), the spontaneous recovery test (middle), and the reinstatement test (right). In the final session of extinction, CS freezing levels did not significantly differ between the groups ($Fs < 2$). By contrast, in the spontaneous recovery test, the CS freezing in Group Gradual was significantly lower than the average level of freezing in the other three groups, ($F_{1,28} = 5.171$, $p = 0.031$, $d = 0.977$, 95% CI $[-1.765, -0.092]$), which did not differ from each other ($Fs < 1$). In the reinstatement test, Group Gradual displayed numerically less CS freezing than the other three groups. However, this difference did not reach significance, and there were no differences among the other three groups.

Taken together, these results indicate that repeated exposure to the lowest intensity shock is not sufficient to protect against fear restoration in a test of spontaneous recovery. Rather, protection is only observed when the shock intensity decreases in a progressive and gradual manner before the shock is omitted altogether. The overall pattern of results was similar in the reinstatement test but did not reach significance due to variability within the groups. Nonetheless, it is clear that the results do not support an account of gradual extinction in terms of a devalued shock representation.

## Experiment 5: A physical context shift attenuates the effectiveness of gradual extinction

According to the *Cochran and Cisler, 2019* model, the size of the prediction error determines how new learning is encoded relative to what is already known. The results of the experiments conducted thus far are consistent with this model, but a feature of both the model's explanation and the previous experiments is that all prediction errors have been generated by a change in the CS-US relationship.

While the model permits contextual (e.g. spatio-temporal) information to be encoded concomitantly with the formation of the CS-US association, it predicts that only prediction errors relating to the CS-US association influence the process of state-splitting. The model also holds that a shift in context, whether it be a change in the physical environment or interpolating a delay between sessions (analogous to a change in the temporal context), means that the most recent active state is no longer most likely to be active (i.e. now all stored states are equally likely to be active). However, such shifts do not influence the process of state-splitting. That is, prediction error produced by a difference in contextual information does not influence whether new information is encoded into an existing or new state. Hence, the model predicts that the gradual extinction effect should survive a physical or temporal context shift: i.e., a physical or temporal context shift will *not* attenuate the effectiveness of gradual extinction in protecting against spontaneous recovery and/or reinstatement. Simulations confirmed these predictions (see *Figure 4B, C, F and G*).

It is worth noting that, within a more general latent state framework (e.g. the models proposed by *Redish et al., 2007*; *Gershman et al., 2017*), prediction errors produced by context changes may well contribute to the process that determines the location of newly encoded information. In that case, the prediction error produced by combining a shift in shock intensity with a shift in physical context would be sufficient to cause state-splitting, resulting in the gradual extinction experience being encoded in a separate state. Thus, from this perspective, shifting the physical context between conditioning and extinction, should not only attenuate the effectiveness of gradual extinction: it should produce the opposite effect, resulting in higher levels of spontaneous recovery relative to the case where there had been no context shift.

Experiment 5 tested these differing predictions by combining gradual extinction with a shift in the physical context. Subjects received conditioning in either context A or context B (see *Figure 4A* for design). Next, rats in both groups received either standard or gradual extinction in context A. Finally, all rats were tested for spontaneous recovery in context A. Thus, all extinction and testing were conducted in the same context, rats only differed on their extinction treatment (gradual or standard) and conditioning context (A - 'same' context as extinction; B - 'different' context to that of extinction).

*Figure 4D* shows mean freezing (±SEM) across the final session of extinction (left) and the spontaneous recovery test (right). One rat from Group Gradual was removed from the analysis due to abnormally high freezing at the test (over 2 SDs when included in the average and 4 SDs when left out). Importantly, the inclusion/removal of this rat did not change the pattern of statistical difference between groups. In the final session of extinction, there was no significant difference in CS freezing between rats exposed to gradual or standard extinction (i.e. extinction type), no significant difference in CS freezing between rats extinguished in the same context as conditioning versus rats extinguished in a different context (i.e. conditioning location), and no significant interaction between these factors ($Fs < 1$). In the spontaneous recovery test, the level of CS freezing did not significantly differ in terms of extinction type or extinction location ($Fs < 2$). However, there was a significant interaction between extinction type and extinction location ($F_{1,28} = 5.827$, $p = 0.025$, $d = 1.260$, $95\% \, CI \, [0.129, 1.578]$). From inspection of *Figure 4D*, the interaction was due to gradual extinction producing less spontaneous recovery than standard extinction when conditioning and extinction occurred in the same context; but more spontaneous recovery than standard extinction when conditioning and extinction occurred in different contexts. Thus, a shift in physical context between conditioning and extinction reversed the advantage of gradual extinction over standard extinction. The implication for latent state models is that combining the reduction in shock intensity with a change in context produced a prediction error sufficiently large to cause state-splitting. Thus, rats exposed to this combination (Group Gradual-Different) exhibited more freezing to the CS at the test than their standardly extinguished counterparts as the CS-shock experiences are likely to have been represented in multiple states of memory. More generally, these results support a view that physical context influences the process of state-splitting, which does not accord with the role afforded to context in the *Cochran and Cisler, 2019* model.

## Experiment 6: A temporal delay attenuates the effectiveness of gradual extinction

The previous experiment showed that a shift in physical context reverses the effectiveness of gradual extinction. This is inconsistent with the *Cochran and Cisler, 2019* model which holds that only

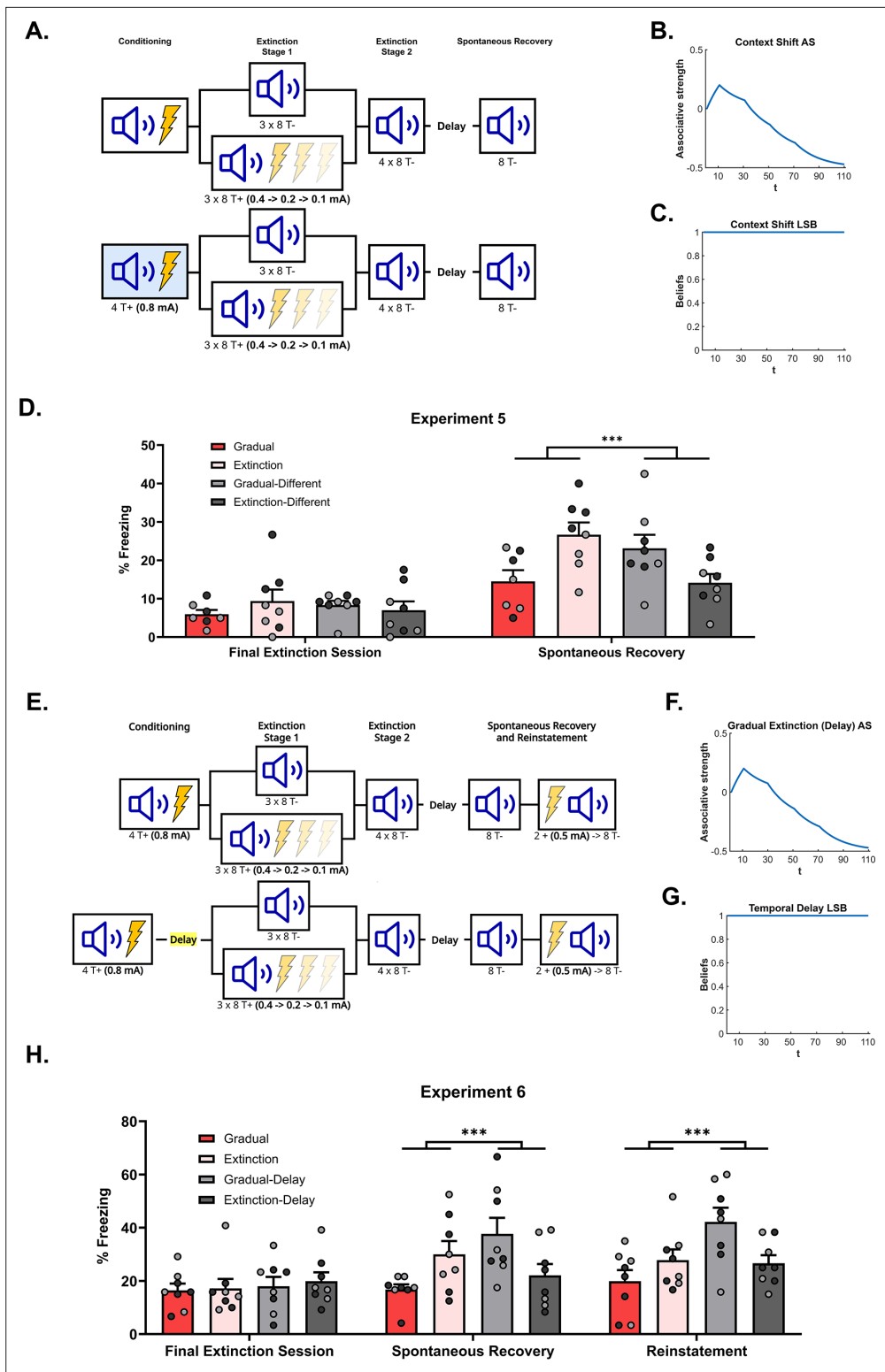

**Figure 4.** The effectiveness of gradual extinction is attenuated by a shift in physical or temporal context. (**A**) Design for Experiment 5: Rats were conditioned with tone-shock pairings in context A (Group Extinction, n=8; Group Gradual, n=8) or in context B (Group Extinction-Different, n=8; Group Gradual-Different, n=8). All groups then received extinction in context A, either standard (Groups Extinction and Extinction-Different) or gradual (Groups Gradual and Gradual-Different) extinction. All groups received further standard extinction before being tested for spontaneous recovery (all in context A). Simulation of the latent state model showing the

*Figure 4 continued on next page*

*Figure 4 continued*

associative strength (**B**) and latent state beliefs (**C**) of Group Gradual-Different across conditioning, extinction, and spontaneous recovery (***Cochran and Cisler, 2019***). The trial structure remained identical to the structure used in the simulation displayed in ***Figure 2C and D*** with the exception of a context shift occurring upon the start of gradual extinction (t=11). The standard extinction and spontaneous recovery trials remained the same. Associative strength (**B**) increases across conditioning, decreases steadily across gradual extinction, and remains low after a time delay. Latent state beliefs (**C**) remain constant across all trials, indicating that a context shift had not induced state-splitting. Thus, the simulations predict that the effectiveness of gradual extinction is not attenuated by a shift in context. (**D**) Percentage freezing levels across the final session of extinction (left) and spontaneous recovery test (right). Bars represent means ± SEM. Dots represent individual freezing levels (males = dark gray, females = light gray). Freezing levels are similar across the final session of extinction for all groups. At test, Group Gradual displayed less conditioned stimulus (CS) freezing than Groups Extinction, however, the pattern was reversed for those who received conditioning in a different context (Group Gradual-Different >Group Extinction-Different). (**E**) Design for Experiment 6: All rats were conditioned with tone-shock pairings and then either received standard extinction or gradual extinction. For half the rats, this occurred on successive days (Group Extinction, n=8; Group Gradual, n=8), while for the other half, these experiences were separated by two weeks (Group Extinction-Delay, n=8; Group Gradual-Delay, n=8). All groups were arranged such that the extinction experiences occurred together. All groups then received further standard extinction and were tested for spontaneous recovery and reinstatement. Simulation of the latent state model showing the associative strength (**F**) and latent state beliefs (**G**) of Group Gradual-Delay across conditioning, extinction, and spontaneous recovery (***Cochran and Cisler, 2019***). The trial structure remained identical to the structure used in the simulation displayed in ***Figure 2C and D*** with the exception of a time delay occurring upon the start of gradual extinction (t=11). The standard extinction and spontaneous recovery trials remained the same. Associative strength (**F**) increased across conditioning, decreased steadily across gradual extinction, and remained low after a time delay. Latent state beliefs (**G**) remained constant across all trials, indicating that a time delay had not induced state-splitting. Thus, the simulations that the effectiveness of gradual extinction is not attenuated by a time delay between conditioning and extinction. (**H**) Percentage freezing levels across the final session of extinction (left) and spontaneous recovery test (right). Bars represent means ± SEM. Dots represent individual freezing levels (males = dark gray, females = light gray). CS freezing levels are similar across the final session of extinction for all groups. At the spontaneous recovery and reinstatement test, Group Gradual displayed less CS freezing than Groups Extinction, however, the pattern was reversed for those who received conditioning in a different context (Group Gradual-Delay >Group Extinction-Delay).

prediction errors relating to the CS-shock relationship affect state-splitting. The aim of this experiment was to determine whether a shift in temporal context also influences the gradual extinction effect. A shift in temporal context was achieved by inserting a delay between conditioning and extinction. The Cochran-Cisler model predicts that, similar to a physical context shift, a temporal shift will reset latent state beliefs but not affect state-splitting. Nonetheless, a physical context shift reversed the effectiveness of gradual extinction; hence, a temporal shift may do the same (for simulations, see ***Figure 4F and G***).

Experiment 6 tested this prediction (design in ***Figure 4E***). Four groups of rats were conditioned to fear a tone CS across its pairings with 0.8 mA shock. Two groups were then subjected to gradual extinction while the remaining two groups received standard extinction. For one group in each of these pairs, the extinction stage commenced one day after conditioning; for the other group, the extinction stage commenced three weeks after conditioning. Importantly, the protocols were arranged so that all rats received extinction at the same time. After extinction, all rats were tested three weeks later for spontaneous recovery and reinstatement of freezing to the CS.

***Figure 4H*** shows mean freezing (±SEM) across the final session of extinction (left), the spontaneous recovery test (middle), and the reinstatement test (right). In the final session of extinction, there was no significant difference in freezing to the CS between rats that received gradual versus standard extinction; no significant difference in freezing to the CS between rats that received extinction one or 21 days after conditioning, and no significant interaction between the factors of extinction type and extinction time ($Fs < 1$). In the spontaneous recovery test, there was no overall difference in CS freezing as a function of extinction type or extinction time ($Fs < 3$). However, there was a significant interaction between these two factors ($F_{1,28} = 10.035$, $p = 0.004$, $d = 2.328$, $95\% \, CI \, [0.396, \, 1.844]$), which was due to the fact that gradual extinction produced less spontaneous recovery than standard

extinction when extinction commenced one day after conditioning, but more spontaneous recovery than standard extinction when extinction commenced three weeks after conditioning.

In the final reinstatement test, averaged across extinction time, there was no overall difference in CS freezing between rats that received either gradual versus standard extinction ($F < 1$). By contrast, averaged across extinction types, the level of CS freezing was greater among rats for which there was a delay between conditioning and extinction compared to those for which there was no delay ($F_{1,28} = 6.270$, $p = 0.01$, $d = 0.902$, $95\% \, CI \, [0.161, 1.610]$). From inspection of *Figure 4H*, this difference is likely due to high freezing in the Gradual-Delay group relative to both non-delay groups. This was confirmed by a significant interaction between the factors of extinction type and extinction time ($F_{1,28} = 7.703$, $p = 0.004$, $d = 2.27$, $95\% \, CI \, [0.257, 1.705]$), which was due to the same pattern of results observed in the spontaneous recovery test. Thus, a shift in temporal context, much like a shift in physical context, reversed the advantage of gradual extinction over standard extinction. This further supports the idea that contextual information contributes to the state-splitting process while not agreeing with the specific role for temporal context suggested by the Cochran-Cisler latent state model.

## Omnibus-analysis of all experiments

The six experiments presented here show that gradual extinction is more effective than standard extinction in protecting against spontaneous recovery and/or reinstatement. Two groups were represented in each of these experiments: Group Gradual and Group Extinction. Across the experiments, these groups differed with respect to the number of extinction sessions that followed the initial three days of extinction (where the groups received CS-US pairings of decreasing intensity or CS-alone presentations, respectively). Moreover, in Experiments 5 and 6 (physical and temporal context shift), the delay to spontaneous recovery testing was three weeks rather than two. Otherwise, the treatment of rats in Groups Gradual and Extinction was identical in each experiment.

The similarity of these two groups across experiments permitted an omnibus analysis comparing all rats that received either gradual or standard extinction. A single Group Gradual (i.e. all rats within Group Gradual in Experiments 1–6) and a single Group Extinction (i.e. all rats within Group Extinction in Experiments 1–6) were formed. An ANOVA assessed freezing levels across the final session of extinction and the spontaneous recovery test. *Figure 5* shows that freezing levels were similar across the final session of extinction, but were higher for Group Extinction compared to Group Gradual across the spontaneous recovery test ($F_{1,92} = 18.554$, $p < 0.001$, $d = 0.891$, $95\% \, CI \, [0.479, 1.298]$).

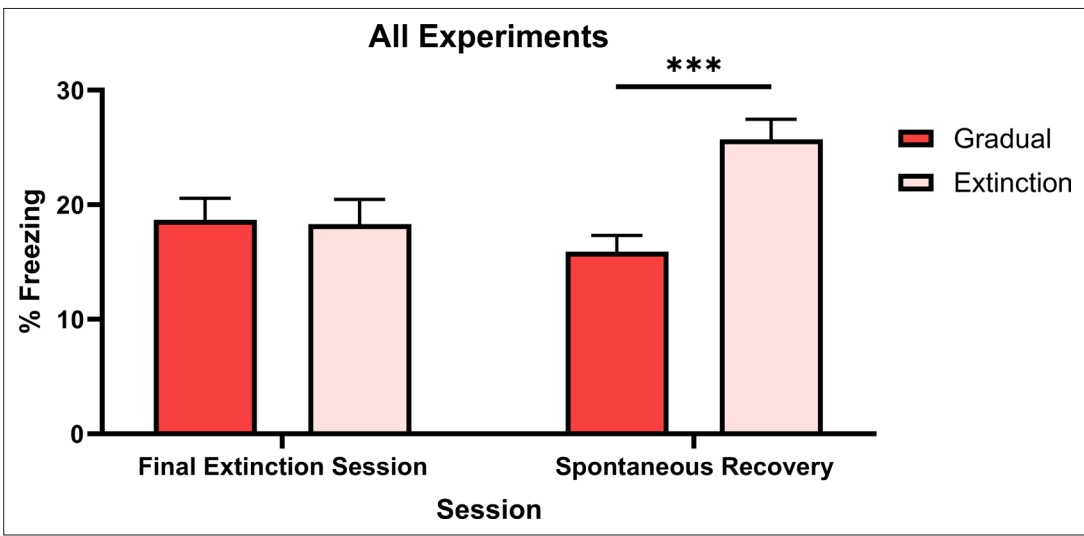

**Figure 5.** Percentage freezing levels across the final session of extinction and spontaneous recovery test for an omnibus analysis of gradual extinction. Group Gradual (n=46) contains data from all rats who received a gradual extinction procedure in any experiment, Group Extinction (n=47) contains data from all rats who received standard extinction in any experiment. Data are means ± SEM.

Averaged across experiments, rats that received gradual or standard extinction displayed similar levels of CS freezing at the final session of extinction, indicating both procedures produced an equivalent decrement in conditioned responding. However, gradually extinguished rats displayed less spontaneous recovery of CS freezing than standardly extinguished rats. Thus, the overall analysis further supports the prediction made by latent state models (e.g. *Cochran and Cisler, 2019*; *Redish et al., 2007*; *Gershman et al., 2010*; *Gershman et al., 2017*): specifically, that gradual extinction is more effective than standard extinction in protecting against spontaneous recovery, as it results in the encoding of conditioning and extinction into a single memory state.

## Discussion

This study examined whether a gradual extinction protocol is more effective than a standard extinction protocol in promoting extinction. In each experiment, rats were first exposed to pairings of a CS and a moderately intense shock US. Over the next few days, rats received additional CS-US pairings; however, the intensity of the shock US was gradually reduced until the CS was presented alone and completely extinguished. Testing revealed that rats subjected to this gradual extinction protocol showed less spontaneous recovery and reinstatement of fear (freezing responses) than rats subjected to a standard extinction protocol (i.e. an abrupt shift from CS-shock pairings to CS alone presentations). Importantly, these effects of gradual extinction depended on the reduction in shock intensity being both progressive and gradual: there were no decreases in spontaneous recovery and/or reinstatement among rats exposed to either the lower-intensity shocks in a pseudo-random order; or abruptly shifted to the lowest intensity shock. The effects of gradual extinction also depended on where and when the US intensity was gradually reduced in relation to the initial CS-US pairings: there were no decreases in spontaneous recovery and/or reinstatement among rats for which the US intensity was reduced in a different context to that of the initial conditioning, or several days after the initial conditioning. Thus, the effects of gradual extinction cannot be attributed to differences in shock exposure per se or the way that shock is represented in memory. Instead, the findings are consistent with accounts of extinction in terms of memory interference: that is, gradually reducing the shock intensity across the shift from conditioning to extinction means that the memory of the CS alone experiences can more effectively interfere with retrieval/expression of the CS-US memory in fear responses.

Latent state models provide a convenient way of thinking about memory interference in extinction, and the latent state model proposed by *Cochran and Cisler, 2019* explains most of the present findings. According to this model, during a standard extinction protocol, omission of the shock US produces a large prediction error; and this large prediction error results in the CS only experiences being encoded separately from the CS-US memories formed in conditioning: i.e., formally, the large prediction error results in state-splitting, such that the CS alone experiences are encoded in a separate state to that of the CS-US experiences in conditioning. This separation means that the CS-US memories are able to be re-expressed in fear responses when the extinguished CS is presented under certain circumstances, including after the passage of time (spontaneous recovery) or following US re-exposure (reinstatement). By contrast, during a gradual extinction protocol, the progressive reductions in US intensity produce a series of small prediction errors which are sufficient to produce new learning but not sufficient to cause state-splitting. Hence, the conditioning and extinction experiences are encoded together in the same state, and the ability of the extinction experiences to interfere with the retrieval/expression of the conditioning experiences is thereby greatly increased.

In this manner, the *Cochran and Cisler, 2019* model accommodates our findings that gradual extinction protects against both spontaneous recovery and reinstatement; and correctly identifies these gradual extinction effects with progressive reductions in US intensity across the shift from conditioning to extinction. This was confirmed in a series of model simulations. When the shock intensity changes progressively and gradually across the shift (i.e. in the gradual extinction protocol), the model predicts that the conditioning and extinction experiences are encoded into a single state of memory, resulting in lower levels of responding when the CS is tested after the lapse of time or following US re-exposure. By contrast, when the shock intensity changes unpredictably from one trial to the next (scrambled control) or is immediately reduced to its lowest possible value (0.1 mA control), the model predicts that the conditioning and extinction experiences are encoded into different memory states, resulting in higher levels of responding when the CS is tested after the lapse of time or following US re-exposure.

It is important to note that, while the *Cochran and Cisler, 2019* model explains the gradual extinction effect, it does not contain any mechanism by which a context change or lapse of time might contribute to the state-splitting process. As such, the model is challenged by the results of Experiments 5 and 6, which showed that the effectiveness of gradual extinction was attenuated when the US intensity was progressively reduced in a different context to that of the initial conditioning. That is, the unexpected CS and shock presentations in a different context attenuated the effectiveness of gradual extinction, suggesting that contextual prediction errors do, indeed, contribute to the state-splitting process. This being said, the impact of a context change on the effectiveness of gradual extinction is *generally* consistent with latent state models. According to the broader class of models, new experiences are encoded into the same state as past experiences when the two are sufficiently similar, and into a new state when they are sufficiently different. Thus, in principle, state-splitting can be identified with any sort of prediction error, including those relating to context. Incorporating a mechanism by which context changes can explicitly affect the state-splitting process would be a useful extension of latent state models, including that proposed by Cochran and Cisler.

The present study adds to a growing body of evidence that manipulations applied across the shift from CS-US pairings to presentations of the CS alone can influence the effectiveness of extinction. For example, *Gershman et al., 2013* and *Bouton et al., 2004* showed that gradually reducing the proportion of reinforced CS presentations results in less spontaneous recovery and slower reacquisition, respectively; though both studies left open fundamental questions about the basis of their findings (see also *Woods and Bouton, 2007*). Similarly, *Popik et al., 2020* demonstrated that, after conditioning with a reasonably intense foot shock, pairing the CS with a very weak foot shock accelerates the loss of fear responses across extinction, and decreases levels of renewal and spontaneous recovery (relative to standard extinction). This procedure, termed 'deconditioning,' resembles the 0.1 mA control condition used in Experiment 4 where rats were conditioned with a strong shock and immediately shifted to the lowest intensity shock prior to a series of CS alone presentations in extinction. In contrast to the findings of Popik and colleagues, we found no evidence that extinction was more effective under these circumstances. However, it is important to note that rats in our 0.1 mA control group were initially conditioned with a strong (0.8 mA) shock before being shifted to the weak shock (0.1 mA), whereas rats in the Popik et al. study were initially conditioned with a moderate (0.5 mA) shock before being shifted to the same weak shock (0.1 mA). According to *Cochran and Cisler, 2019* and other latent state models, rats in our 0.1 mA control group will have experienced a greater prediction error across the shift from conditioning to extinction compared to rats in the deconditioning protocol (*Popik et al., 2020*). Hence, they are more likely to have encoded the CS alone experiences in a different state to that of the initial conditioning experiences, resulting in normal levels of spontaneous recovery and reinstatement (i.e. no evidence for deconditioning). That is, we take our results and those of Popik et al. to provide very strong evidence that the size of the prediction error across the shift from conditioning to extinction determines the effectiveness of the latter; specifically, a series of small errors are more effective than a moderate or large error in promoting extinction.

One question that remains to be addressed concerns how state-splitting signals are registered in the brain. Some evidence suggests that they may be registered in activity of the adrenergic system. For example, neurons in the locus coeruleus respond to contingency changes across a variety of learning procedures (*Devauges and Sara, 1990*; *Sales et al., 2019*; *Vankov et al., 1995*) including omission of an expected outcome in extinction (*Sara et al., 1994*). These neurons also provide input to the hippocampus, resulting in the suppression of previously activated neuronal ensembles and the recruitment of new ensembles (*Grella et al., 2019*; *Sanders et al., 2020*). These and other findings provide a framework for thinking about extinction in the standard and gradual protocols. In the standard extinction protocol, the large prediction error produced by shock omission activates noradrenergic neurons in the locus coeruleus; and this, in turn, provides a signal for state-splitting in the hippocampus. Hence, the CS alone experiences are encoded separately from the prior CS-US experiences, and the latter retain their capacity to be re-expressed in behavior. By contrast, in the gradual extinction protocol, the small prediction errors produced by progressive reductions in US intensity are not large enough to activate neurons in the locus coeruleus. Hence, the CS alone experiences are encoded together with the prior CS-US experiences and, thereby, more effectively interfere with retrieval/expression of the latter (for similar arguments in relation to memory reconsolidation, see *Bonanno et al., 2023*; *Ferrara et al., 2019*; *Popik et al., 2023*).

In summary, the present study has shown that a gradual extinction protocol is more effective than a standard extinction protocol in promoting extinction. That is, gradually reducing the US intensity across a series of CS-US pairings increased the effectiveness of extinction, which was evident in lower levels of spontaneous recovery and reinstatement. These findings are consistent with latent state models, including tthose proposed by *Cochran and Cisler, 2019*. According to these models, prediction error determines how new experiences are encoded in states of memory. Small prediction errors result in new experiences being encoded into the same memory state as past experiences, which creates interference in recall of either experience. By contrast, large prediction errors result in new experiences being encoded into a new memory state, which minimizes interference when either experience is recalled. Future work will examine how the brain uses prediction error signals to update existing memory states (small errors) or create new ones (large errors), including the role of the central adrenergic system in mediating these different effects.

## Materials and methods
### Experiment 1
### Subjects
The subjects were 16 experimentally naïve, adult male Long Evans rats, weighing between 250 and 400 g. They were obtained from the colony maintained by the Biological Resources Centre at the University of New South Wales and housed by sex in plastic tubs (67 cm length × 40 cm width × 22 cm height) with four rats per tub. Food and water were available ad libitum across the experiment. The tubs were located in a colony room maintained at 21 degrees Celsius and kept on a 12:12 hr light:dark cycle with lights on at 0700. Rats were handled daily for at least 5 days prior to the beginning of the experiment. The procedures were approved by the Animal Ethics Committee of The University of New South Wales.

### Apparatus
The apparatus consisted of a set of four identical chambers. Each chamber was 23 cm long × 21 cm deep × 23 cm high. The front, rear, and side walls were clear Plexiglass, and the side walls were aluminum. The floor consisted in stainless-steel rods, 2 mm in diameter and spaced 10 mm apart. A shock could be delivered to the floor of each chamber via a custom-built generator located in another room in the laboratory. Each chamber was located in a sound- and light-attenuating wooden cabinet. A speaker and a 2 × 3 array of light-emitting diodes (LEDs) were mounted on the back wall of each cabinet. A camera mounted on the back wall of the cabinet was used to record the activity of each rat, and each chamber was illuminated by an infrared light to permit observation of each rat. The camera was connected to a monitor and DVD recorder located in another room of the laboratory.

### Stimuli
The conditioned stimulus was a 1000 Hz 70 dB tone and the US was a 0.5 s foot shock. The intensity of the foot shock was 0.8 mA in conditioning and ranged between 0.4, 0.2, and 0.1 mA during gradual extinction. Stimuli were programmed and presented using Matlab software.

### Procedure
#### Context exposure
On days 1 and 2 all rats were exposed to the context where training and testing occurred. There were two 20 min sessions each day, one in the morning and the other approximately 3 hr later in the afternoon.

#### Conditioning
On day 3, all rats received four tone-shock pairings, where each 30 s tone presentation co-terminated with a 0.5 s 0.8 mA foot shock. For the current and subsequent experiments, across conditioning, extinction stage 1, extinction stage 2, and testing, the first tone presentation occurred 3 min after the rat was placed in the context, the inter-trial interval was 3 min and rats remained in the chamber for 2 min after the final scheduled event (tone or shock) before being returned to their home cage.

### Extinction stage 1

Rats were randomly allocated to two groups (n=8, each composed of four males and four females). On each of days 4–6, rats in Group Standard Extinction (E) received eight 30 s tone-alone presentations. On each of these days, rats in Group Gradual Extinction (G) received eight tone-shock pairings. The pairings differed from conditioning in that the shock intensity was decreased between days. Specifically, on day 4, the intensity was 0.4 mA, on day 5, 0.2 mA, and on day 6, 0.1 mA.

### Extinction stage 2

On each of days 7 and 8, all groups received standard extinction consisting in eight tone-alone presentations.

### Spontaneous recovery

On days 9–22, the rats remained in their home tubs in the colony room. On day 23, all rats received a spontaneous recovery test consisting of eight tone-alone presentations.

### Statistical analysis

Freezing was the measure of conditioned fear. Freezing was defined as the absence of all movements except those related to breathing (*Fanselow, 1980*). A time sampling procedure was used in which each rat was scored as freezing or not freezing every 2 s during the 30 s of each CS presentation and the 30 s immediately prior to each presentation. A percentage score was calculated for each rat, as the proportion of freezing observations to total observations. Freezing data were scored by the experimenter and an experienced observer who was blind to the experimental manipulations. The correlation between the two sets of scores was greater than 0.9 but any discrepancies were resolved in favor of the naïve observer.

Conditioning, extinction, and spontaneous recovery data were analyzed using a mixed-model ANOVA with a between the factor of Group and a within a factor of Day (for Extinction) or Trial (for Conditioning and Spontaneous Recovery). The type 1 error rate was controlled at $\alpha=0.05$ which gave an F critical of 4.7 (df 1 and 13). Standardized 95% confidence intervals (CIs) were reported for significant results. Cohen's d (0.8=large) and Partial Eta Squared ($\eta_p^2$ (0.14=large) were reported as a measure of effect size for between and within-factor results, respectively.

## Experiment 2

### Subjects

The subjects were 24 (12 male and 12 female) experimentally naïve rats of the same age, source, and maintained under the conditions described previously. The apparatus and stimuli were those used previously.

### Procedure

#### Context exposure and conditioning

On days 1–3, all rats received context exposure and conditioning in the manner described in Experiment 1.

#### Extinction stage 1

Rats were randomly allocated to three groups (n=8). On each of days 4–6, rats in Groups Extinction and Gradual Extinction received the same treatments as the corresponding groups in Experiment 1, while those in Group Single Extinction were handled but remained in their home tubs in the colony room.

#### Extinction stage 2

On day 7, all groups received eight tone alone presentations.

#### Spontaneous recovery and reinstatement

On day 21 all rats received a spontaneous recovery test in the manner described previously. On day 22, all rats received two 0.5 s 0.5 mA foot shocks. The first shock occurred 2 min after placement in

in the context and the second shock occurred 5 s later. Rats remained in the context for 3 min after the shock before being returned to their home tubs in the colony room. Approximately, 5 min later, the rats were returned to the conditioning chambers and tested with eight tonealone presentations.

## Statistical analysis

The data from conditioning, the common extinction session, the spontaneous recovery, and reinstatement tests were analyzed using a set of planned orthogonal contrasts (*Hays, 1967*). Data from the gradual extinction and extinction sessions (days 4–6) were analyzed using a mixed-model ANOVA with a between factor of Group and a within factor of Day. The type one error rate was controlled at $\alpha$=0.05 which gave an F critical of 4.600 (df 1 and 14) for extinction training and 4.325 for all other stages (1 and 21 df). The first of the planned contrasts compared Groups Extinction and Gradual Extinction to Group Single Extinction to assess whether the treatments accorded the former groups had produced a greater long-term depressive effect on conditioned responding than the latter, i.e., the group just given the single extinction session. The second contrast compared Group Gradual Extinction to Group Extinction to assess whether the former exhibited less fear recovery on both tests than the latter, as had been found in the previous experiment for the spontaneous recovery test.

## Experiment 3

### Subjects, apparatus, and stimuli

The subjects were 12 male and 12 female experimentally naïve rats of the same age, from the same source and maintained under the same conditions described previously. The apparatus and stimuli were those used previously.

### Procedure

#### Context exposure and conditioning

On days 1–3, all rats received context exposure and conditioning in the manner described in Experiment 1.

#### Extinction stage 1

Rats were randomly allocated to three groups (n=8). On each of days 4–6, rats in group extinction and gradual extinction received tone alone presentations and progressive reductions in shock intensity, respectively, in the manner described in Experiment 1. On each of these days, Rats in Group Scrambled continued to receive pairings of the tone and shock. The number and intensity of the pairings matched those administered Group Gradual Extinction, but were arranged in a pseudo-random order. The order is as follows: Day 1=0.4, 0.1, 0.2, 0.2, 0.4, 0.1, 0.4, 0.2 mA; Day 2=0.2, 0.1, 0.4, 0.1, 0.4, 0.2, 0.1; Day 3=0.1, 0.4, 0.2, 0.4, 0.1, 0.1, 0.4, 0.2. All subjects within the group received the same sequence.

#### Extinction stage 2

On each of days 7–12, all rats received eight tone alone presentations in the manner described previously.

#### Spontaneous recovery and reinstatement

On day 26, all rats received a spontaneous recovery test in the manner described in Experiment 1, and on day 27, a reinstatement test in the manner described in Experiment 2.

### Statistical analysis

All data were analyzed using a set of planned orthogonal contrasts (*Hays, 1967*). The type one error rate was controlled at $\alpha$=0.05 which gave an F critical of 4.325 (1 and 21 df) for all stages. The first contrast compared Group Gradual to Groups Extinction and Scrambled to assess whether the progressive reduction in shock intensity produced less fear restoration than extinction throughout or extinction preceded by the scrambled shock protocol. The second contrast compared Group Extinction against Group Scrambled to assess whether the reduction in shock intensity per se produced less fear restoration than standard extinction.

## Experiment 4

### Subjects, apparatus, and stimuli

The subjects were 16 male and 16 female experimentally naïve rats of the same age, from the same source and maintained under the same conditions described previously. The apparatus and stimuli were those used previously.

### Procedure

#### Context exposure and conditioning

On days 1–3, all rats received context exposure and conditioning in the manner described in Experiment 1.

#### Extinction stage 1

Rats were randomly allocated to four groups (n=8). On each of days 4–6, rats in Groups Extinction and Gradual Extinction received the treatments described in Experiment 1. On each of these days, rats in Group 0.1 × 3 received pairings of the tone and a 0.1 mA shock, while those in Group 0.1 received pairings of the tone and the 0.1 mA shock on day 4 followed by tone alone presentations on days 5 and 6.

#### Extinction stage 2

On each of days 7–9, all rats received eight tone-alone presentations.

#### Spontaneous recovery and reinstatement

On day 30, all rats received a spontaneous recovery test in the manner described in Experiment 1, and on day 31, a reinstatement test in the manner described in Experiment 2.

### Statistical analysis

All data were analyzed using a set of planned orthogonal contrasts (*Hays, 1967*). The type one error rate was controlled at $\alpha$=0.05 which gave an F critical of 4.196 for all stages (1 and 28 df).

The first contrast compared Group Gradual extinction against the remaining three groups to assess whether the progressive reduction in shock intensity produced less fear restoration than regular extinction or the low shock intensity manipulations. The second contrast compared Groups 0.1 × 3 and 0.1 against Group Standard Extinction to assess whether receiving the lower-intensity shocks produced less fear restoration than extinction. The final contrast compared Group 0.1 × 3 against Group 0.1 to assess whether receiving the low-intensity shocks for a single day differed in its fear restoration consequences than three such days.

## Experiment 5

### Subjects

The subjects were 16 male and 16 female, experimentally naive rats of the same age, from the same source and maintained under the same conditions described previously.

Apparatus and stimuli. Context A remained identical to the context used in previous experiments. Context B differed. Each chamber was 30 cm long × 26 cm deep × 30 cm high. The front and rear walls were clear Plexiglas, and the side walls were aluminum. The floor was made of stainless-steel rods, 7 mm in diameter, and spaced 18 mm apart. An almond scent was also present in the chambers to further differentiate the two contexts.

### Procedure

#### Context exposure

Rats were randomly allocated to four groups (n=8). On days 1–2, all rats received a single 20 min exposure to Context A and a 20 min exposure to Context B. The exposures were separated by 3 hr and the order of the exposures was counterbalanced within groups. The order of exposures on day 1 was reversed on day 2, such that if the order on day 1 was A then B, the order on day 2 was B then A, and if B then A on day 1, it was A then B on day 2.

## Conditioning

On day 3, rats in Groups Extinction and Gradual Extinction received conditioning, as described previously, in context A, while those in Groups Extinction Different and Gradual Extinction Different received conditioning in context B. All rats were also exposed to the other context, such that if conditioned in context A, they were exposed for an equal amount of time to context B and vice versa. The order of exposures to the two contexts was counterbalanced within groups.

## Extinctionstage 1

On each of days 4–6, all rats received extinction in context A. Groups Extinction and Extinction Different received standard extinction as described for Group Extinction in previous experiments. Rats in Groups Gradual Extinction and Gradual Extinction Different continued to receive pairings of the tone and shock in the manner described for Group Gradual Extinction in previous experiments.

## Extinction stage 2

On each of days 7–10, all groups received eight tone alone presentations.

## Spontaneous recovery

On day 31, all rats received a spontaneous recovery test in context A, in the manner described in Experiment 1.

## Statistical analysis

All data were analyzed using a 2 × 2 mixed-model ANOVA with between factors of context and type of extinction. A within factor of trial for conditioning and testing and of day for extinction was included. The type one error rate was controlled at $\alpha=0.05$ which gave an F critical of 4.196 for all stages (1 and 28 df).

## Experiment 6

### Subjects, apparatus, and stimuli

The subjects were 16 male and 16 female experimentally naïve rats of the same age, from the same source and maintained under the same conditions described previously. The apparatus and stimuli were those used previously.

### Procedure

#### Context exposure and Cconditioning

Rats were randomly allocated to four groups (n=8). On days 1–3, rats in Groups Extinction-Delay (E-D) and Gradual-Delay (G-D) received context exposure and conditioning in the manner described in Experiment 1. They remained in their home cage on days 4–21. On days 19–21, rats in Groups Extinction and Gradual received context exposure and conditioning in the manner described in Experiment 1.

#### Extinction stage 1

On each of days 22–24, rats in Groups Extinction and Extinction-Delay received standard extinction as described in Experiment 1, and Groups Gradual Extinction and Gradual-Delay received gradual extinction as described in Experiment 1.

#### Extinction stage 2

On each of days 25–28, all rats received eight tone alone presentations.

#### Spontaneous recovery and reinstatement

On day 59, all rats received a spontaneous recovery test in the manner described in Experiment 1, and on day 60, a reinstatement test in the manner described in Experiment 2.

## Statistical analysis

All data were analyzed using a 2x2 mixed-model ANOVA with between factors of delay and type of extinction. A within factor of trial for conditioning and testing, and of day for extinction was included. The type one error rate was controlled at $\alpha=0.05$ which gave an F critical of 4.196 for all stages (1 and 28 df).

## Acknowledgements

This work was supported by an Australian Government Research Training Fellowship to NGWK, an Australian Research Council (ARC) Discovery Early Career Researcher Award to JCL (DE210100292), an ARC Discovery Grant to RFW (DP2201036501) and an ARC Future Fellowship to NMH (FT190100697).

## Additional information

### Funding

| Funder | Grant reference number | Author |
| --- | --- | --- |
| Australian Research Council | DP2201036501 | Simon Killcross<br>R Fred Westbrook<br>Nathan M Holmes |
| Australian Research Council | FT190100697 | Nathan M Holmes |
| Australian Research Council | DE210100292 | Jessica C Lee |
| Australian Government | | Nicholas GW Kennedy |

The funders had no role in study design, data collection and interpretation, or the decision to submit the work for publication.

### Author contributions

Nicholas GW Kennedy, Conceptualization, Formal analysis, Investigation, Methodology, Writing – original draft, Writing – review and editing; Jessica C Lee, Conceptualization, Supervision, Writing – review and editing; Simon Killcross, Conceptualization, Funding acquisition, Writing – review and editing; R Fred Westbrook, Conceptualization, Supervision, Funding acquisition, Writing – review and editing; Nathan M Holmes, Conceptualization, Supervision, Funding acquisition, Project administration, Writing – review and editing

### Author ORCIDs

Nicholas GW Kennedy https://orcid.org/0000-0002-1585-8240
Nathan M Holmes https://orcid.org/0000-0002-0592-2026

### Ethics

This study was performed in strict accordance with the recommendations in the Guide for the Care and Use of Laboratory Animals by the National Health and Medical Research Council of Australia. All of the animals were handled according to approved institutional Animal Care and Ethics Committee (ACEC) protocols of the University of New South Wales. The protocol was approved by the UNSW ACEC (Permit Number: 21-132B).

Reviewer #1 (Public Review): https://doi.org/10.7554/eLife.95849.3.sa1
Reviewer #2 (Public Review): https://doi.org/10.7554/eLife.95849.3.sa2
Reviewer #3 (Public Review): https://doi.org/10.7554/eLife.95849.3.sa3
Author response https://doi.org/10.7554/eLife.95849.3.sa4

## Additional files

### Supplementary files
• MDAR checklist

### Data availability
All data generated or analyzed during this study are included in the manuscript and supporting files. Source data files are provided via the Open Science Framework repository.

The following dataset was generated:

| Author(s) | Year | Dataset title | Dataset URL | Database and Identifier |
|---|---|---|---|---|
| Kennedy N | 2024 | Prediction error determines how memories are organized in the brain | https://osf.io/5D9Q3/ | Open Science Framework, 5D9Q3 |

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

## Appendix 1

### Analysis of training data

#### Experiment 1

Conditioning and extinction were successful. *Appendix 1—figure 1A* displays mean freezing (±SEM) across conditioning (left panel) and extinction (middle and right panels). Averaged across groups, CS freezing levels increased across conditioning trials ($F_{1,13} = 47.884$, $p < 0.001$, $\eta_p^2 = 0.786$, $95\% \, CI \, [1.297, 2.474]$) and decreased across extinction sessions ($F_{1,13} = 178.863$, $p < 0.001$, $\eta_p^2 = 0.931$, $95\% \, CI \, [-2.576, -1.859]$). There was no significant difference between the two groups in the rate at which freezing changed across conditioning trials or extinction sessions, or in the overall levels of freezing in each stage (*F*s <1).

#### Experiment 2

Conditioning and extinction were successful. *Appendix 1—figure 1B* displays mean freezing (±SEM) across conditioning (left panel) and extinction (middle and right panels). Averaged across groups, CS freezing levels increased across conditioning trials ($F_{1,21} = 93.575$, $p < 0.001$, $\eta_p^2 = 0.815$, $95\% \, CI \, [1.469, 2.274]$). There was no significant difference between the groups in the rate at which freezing increased, or in the overall level of freezing (*F*s <1).

Averaged across Groups, CS freezing levels did not differ across the first three days of extinction (*F*<1). There was no significant difference between the two groups in overall freezing across these days (*F*<1). However, there was a significant linear x-group interaction ($F_{1,21} = 6.972$, $p = 0.016$, $\eta_p^2 = 0.247$, $95\% \, CI \, [0.217, 2.091]$), which was due to the fact that freezing decreased across sessions for Group Extinction and increased across sessions for Group Gradual. The increase in freezing in Group Gradual is likely reflects a decrease in escape behaviours as the intensity of the US was decreased.

#### Experiment 3

Due to computer malfunction, freezing data for conditioning and the first eight extinction sessions (out of nine total) were lost.

#### Experiment 4

Conditioning and extinction were successful. *Appendix 1—figure 1C* displays mean freezing (±SEM) across conditioning (left panel) and extinction (middle and right panels). Averaged across groups, CS freezing increased across conditioning trials ($F_{1,28} = 122.465$, $p < 0.001$, $\eta_p^2 = 0.813$, $95\% \, CI \, [1.250, 1.818]$). Group Gradual displayed less overall freezing ($F_{1,28} = 8.775$, $p = 0.006$, $d = 1.20$, $95\% \, CI \, [0.198, 1.085]$) and a slower rate of increase of freezing($F_{1,28} = 4.424$, $p = 0.045$, $\eta_p^2 = 0.137$, $95\% \, CI \, [0.018, 1.329]$) compared to the remaining three groups. Moreover, the rate of freezing increase was slower for Group 0.1 × 3 relative to Group 0.1 × 1 ($F_{1,28} = 8.084$, $p = 0.045$, $\eta_p^2 = 0.224$, $95\% \, CI \, [-1.918, -0.312]$). Inspection of *Appendix 1—figure 1C* (left panel) reveals that these differences are likely driven by low freezing on Trial 4 for Group Gradual and Group 0.1 × 3 respectively. Rats were displaying higher levels of escape behaviors, thus, less freezing was observed.

Averaged across groups, CS freezing decreased across extinction sessions ($F_{1,28} = 50.120$, $p < 0.001$, $\eta_p^2 = 0.641$, $95\% \, CI \, [-1.643, -0.905]$). Group gradually displayed a lesser decrease in CS freezing across sessions relative to the remaining three groups ($F_{1,28} = 4.539$, $p = 0.042$, $\eta_p^2 = 0.139$, $95\% \, CI \, [-1.737, -0.034]$) which is likely due to a lower level of initial freezing in Group Gradual. There were no between-group differences in overall freezing across the extinction sessions.

#### Experiment 5

Conditioning and extinction were successful. *Appendix 1—figure 2A* displays mean freezing (±SEM) across conditioning (left panel) and extinction (middle and right panels). Averaged across groups, CS freezing increased across conditioning trials ($F_{1,28} = 64.865$, $p < 0.001$, $\eta_p^2 = 0.698$, $95\% \, CI \, [1.150, 1.935]$). There was no significant difference between the groups in the rate at which freezing increased, or in the overall level of freezing (*F*s <2). Averaged across groups, CS freezing decreased across extinction sessions ($F_{1,28} = 170.038$, $p < 0.001$, $\eta_p^2 = 0.858$, $95\% \, CI \, [-2.473, -1.801]$). Moreover, the

rate at which freezing decreased was significantly different for rats that received gradual or standard extinction $(F_{1,28} = 8.610, p = 0.007, \eta_p^2 = 0.234, 95\% \, CI \, [-1.633, -0.290])$. This difference reflects an initial maintenance of freezing in the groups receiving gradual extinction as compared to an initial decrease in the groups receiving standard extinction. All other main effects and interactions were not significant $(Fs <4)$.

## Experiment 6

Conditioning and extinction were successful. ***Appendix 1—figure 2B*** displays mean freezing (±SEM) across conditioning (left panel) and extinction (middle and right panels). Averaged across groups, the level of freezing increased across conditioning trials $(F_{1,28} = 90.470, p < 0.001, \eta_p^2 = 0.786, 95\% \, CI \, [1.358, 2.104])$ and decreased across sessions of extinction $(F_{1,28} = 183.720, p < 0.001, \eta_p^2 = 0.867, 95\% \, CI \, [-2.288, -1.687])$. The rate at which freezing increased across conditioning and decreased across extinction did not differ between the groups, and there was no significant difference between the groups in the overall levels of freezing in either stage $(Fs<4)$.

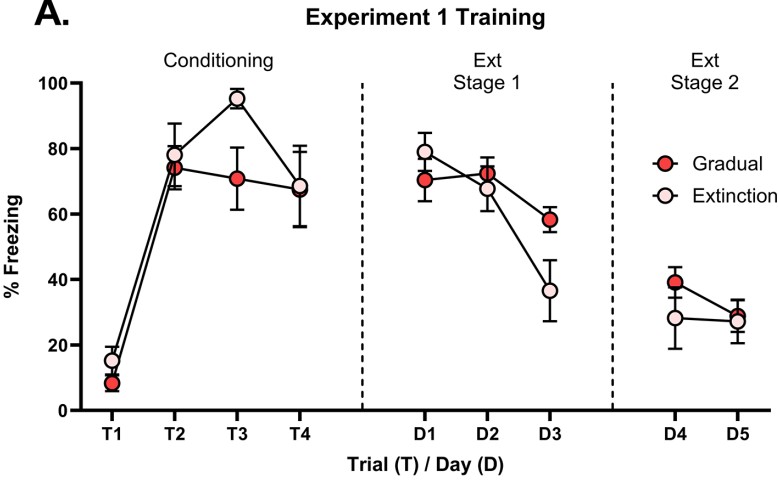

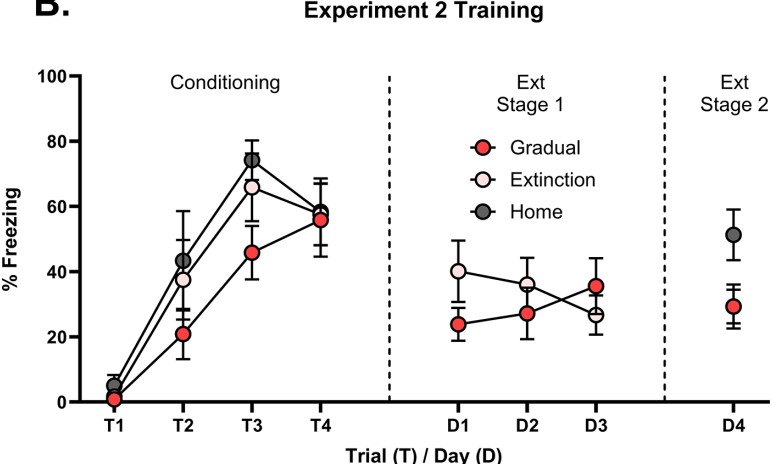

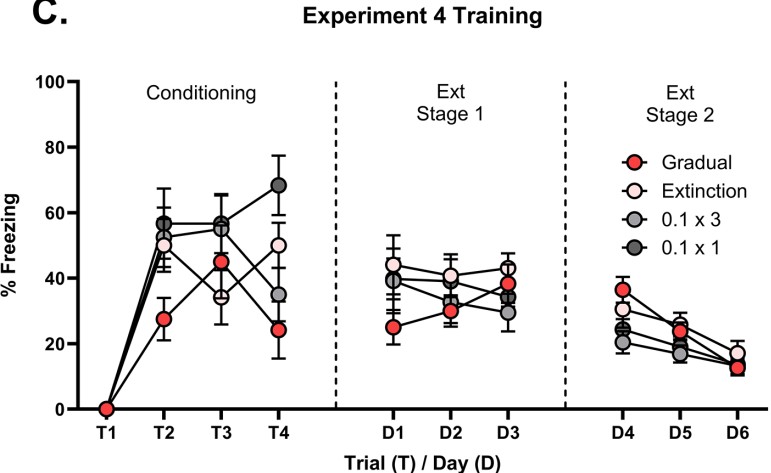

**Appendix 1—figure 1.** Percentage conditioned stimulus (CS) freezing (±SEM) across conditioning and extinction for Experiments 1, 2, and 4. Extinction is split into extinction stage 1 and stage 2. The first stage of extinction refers to extinction sessions where groups were receiving different treatments (i.e. CS alone or continued CS-US pairings). The second stage of extinction is where all groups were receiving CS alone presentations. (**A**) CS Freezing increased across conditioning trials (left) and decreased across extinction stage 1 (middle) and extinction stage 2 (right) at similar rates for both gradual and standard extinction groups. (**B**) CS freezing increased across conditioning trials (left) at similar rates for both groups. Group Gradual displayed an increase in freezing across

*Appendix 1—figure 1 continued on next page*

*Appendix 1—figure 1 continued*

extinction stage 1 while Group Extinction displayed a decrease. Both groups displayed similar levels of CS freezing at extinction stage 2, which appeared to be less than Group No Extinction. (**C**) CS freezing increased across conditioning trials (left) for all groups, however, Group Gradual appears to have increased less than the other three groups. Group Gradual again shows an increase in freezing across extinction stage 1 (middle) compared to decreases in the remaining three groups. Groups decreased at similar rates across extinction stage 2 (right).

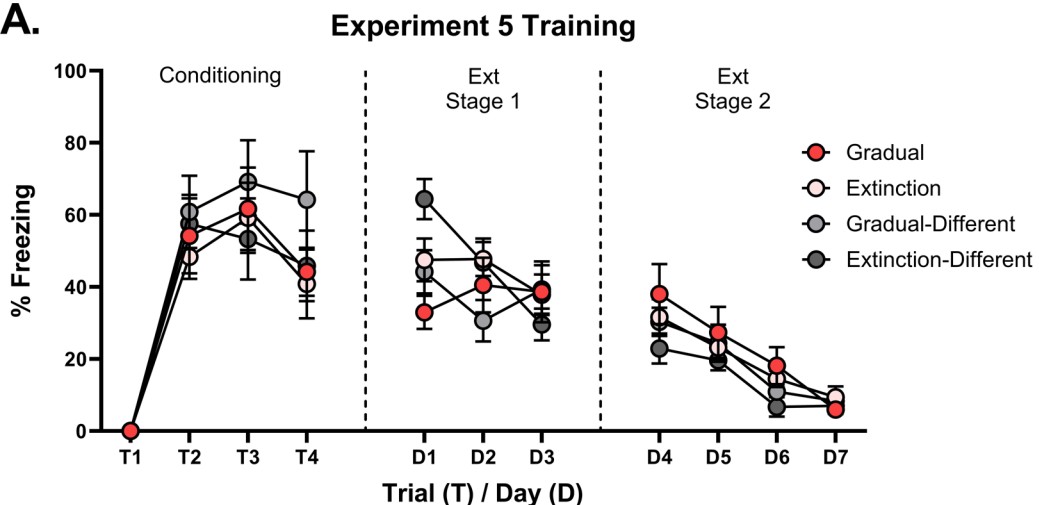

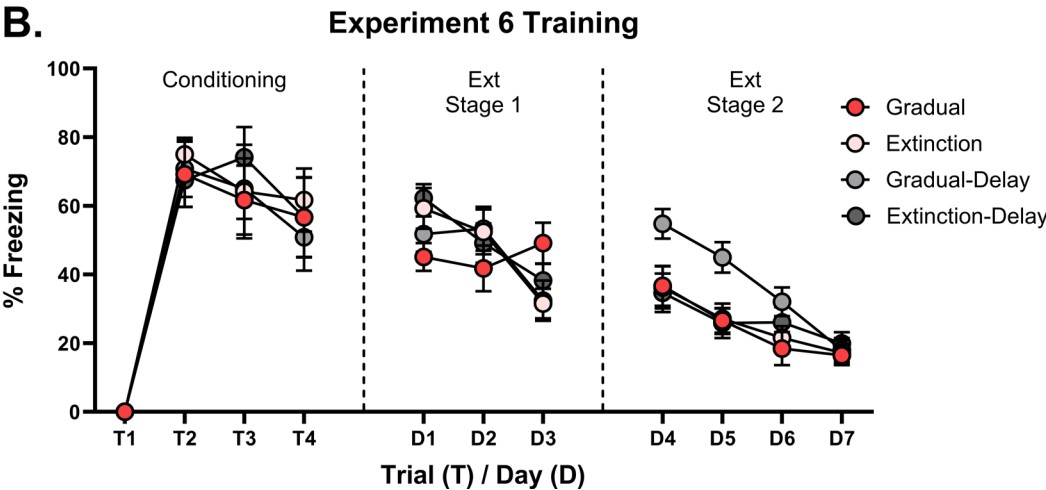

**Appendix 1—figure 2.** Percentage conditioned stimulus (CS) freezing (±SEM) across conditioning and extinction for Experiments 5 and 6. Extinction is split into extinction stage 1 and stage 2. The first stage of extinction refers to extinction sessions where groups were receiving different treatments (i.e. CS alone or continued CS-US pairings). The second stage of extinction is where all groups were receiving CS alone presentations. (**A**) Freezing increased across conditioning trials (left) at similar rates for all groups. Freezing decreased across extinction trials and at a greater rate for those who received standard extinction compared to gradual extinction (regardless of context) across stages 1 and 2 of extinction. (**B**) Freezing increased across conditioning (left) and decreased across extinction stages 1 (middle) and 2 (right) at similar rates for all groups.

## Appendix 2

### Spontaneous recovery and reinstatement trial-by-trial data

*Appendix 2—figure 1* displays trial-by-trial data from the spontaneous recovery and reinstatement tests (where appropriate) from Experiments 1–4 and *Appendix 2—figure 2* displays the same data from Experiments 5 and 6. Please note that the source data files for all experiments conducted in this study are available via the Open Science Framework repository: https://osf.io/5D9Q3/.

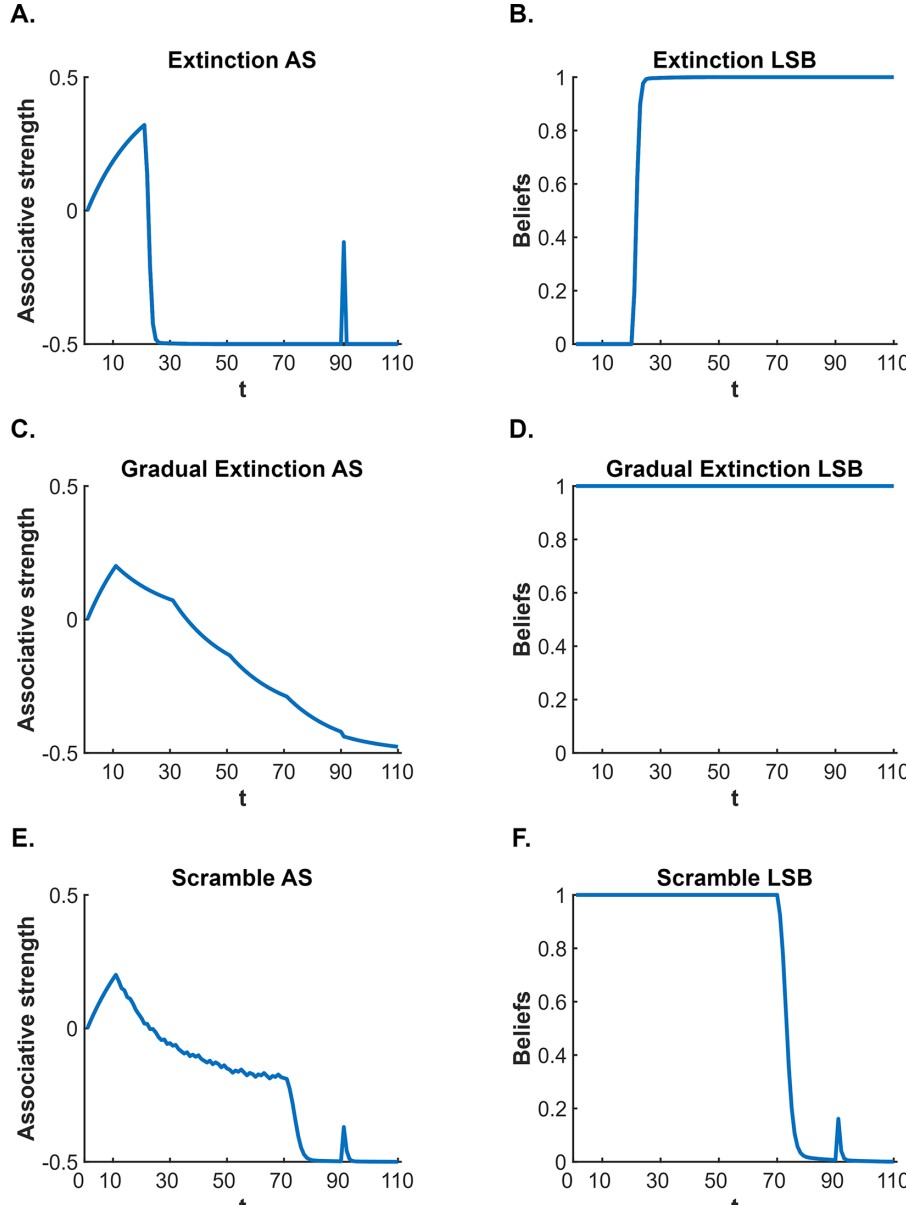

**Appendix 2—figure 1.** Percentage conditioned stimulus (CS) freezing ( ± SEM) across trials for spontaneous recovery and reinstatement tests for Experiments 1–4. Percentage freezing levels across the spontaneous recovery (**A**) for experiment 1, spontaneous recovery (**B**) and reinstatement test (**C**) for experiment 2, spontaneous recovery (**D**) and reinstatement test (**E**) for experiment 3 and spontaneous recovery (**F**) and reinstatement test (**G**) for experiment 4.

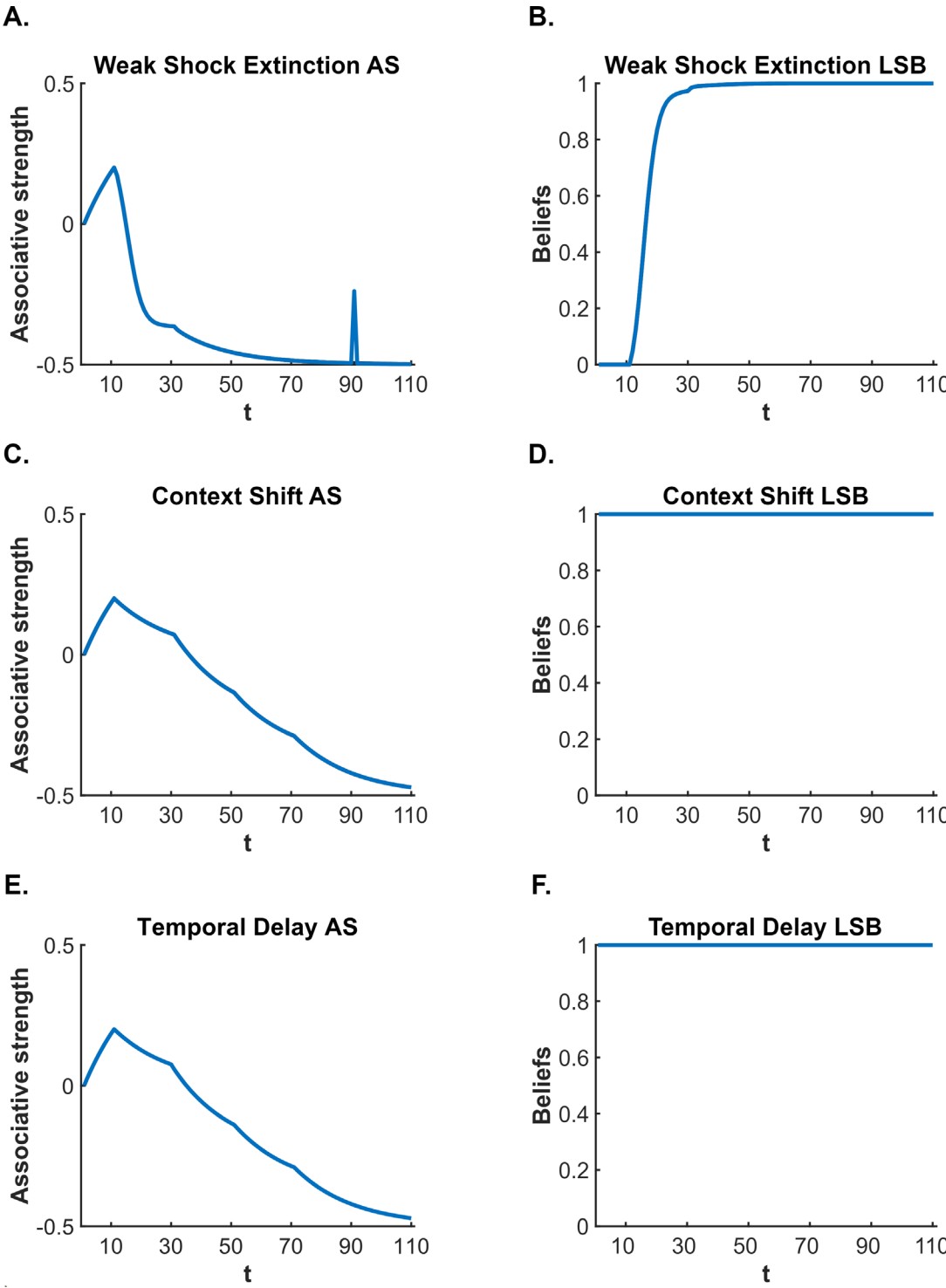

**Appendix 2—figure 2.** Percentage conditioned stimulus (CS) freezing ( ± SEM) across trials for spontaneous recovery and reinstatement tests for Experiments 5–6. Percentage freezing levels across the spontaneous recovery (**A**) for experiment 5 and spontaneous recovery (**B**) and reinstatement test (**C**) for experiment 6.

# Appendix 3

## Simulations of Cochran and Cisler's (2019) model

### Brief model description

*Cochran and Cisler, 2019* start with the assumption that a learning agent wants to predict rewards in their environment. The agent believes that rewards can be predicted as a function of the environmental cues present, a latent state, and a latent error. A latent state indicates which distinct cue-reward contingencies are active in the environment. Latent error refers to the variability in reward generation. To predict rewards, the agent must invert their world view to identify which latent state is currently active, the contingencies between cues and rewards for that latent state, and the variability in rewards associated with that state. For a full characterization of the model, as well as a mathematical justification under the premise of computational rationality (*Gershman et al., 2015*) see *Cochran and Cisler, 2019*.

### Parameters

The parameters chosen (*Appendix 3—table 1*) were those used by *Cochran and Cisler, 2019* to simulate an array of classical conditioning phenomena. Their simulations were resistant to variations in each individual parameter of ± 10% see S1 Text; *Cochran and Cisler, 2019*.

### General simulation details

All simulations followed the same general structure: a conditioning stage, an extinction stage, and a spontaneous recovery stage. The number of trials (t) per stage was consistent across the different simulations. The conditioning stage consisted of 10 trials where the CS was paired with a reward (R). The extinction stage consisted of 80 trials where the CS was presented alone or paired with a weaker version of the reward before being presented alone. Finally, the spontaneous recovery stage consisted of 20 trials where the CS was presented alone after a time-delay. The conditioning and spontaneous recovery stages remained identical across simulations; however, the extinction stage differed between simulations.

For each simulation, we present the associative strength of the CS across its presentations as well as inferred beliefs about which latent state is active. Associative strength was defined as the expected reward conditional on the cue being presented alone. Belief in a latent state was defined as the estimated probability of the latent state given observations.

### Standard extinction

*Appendix 3—figure 1A, B* display simulations in relation to a standard extinction protocol. Extinction was simulated by pairing a CS with reward (Conditioning, t=1–10, R=1) and then presenting that CS in the absence of reward (Extinction, t=11–90, R=0). Finally, after a time delay (the equivalent of 200 trials in the model), the CS again is presented in the absence of the reward (Spontaneous Recovery, t=91–110, R=0). Associative strength increases across conditioning and declines across extinction before briefly recovering at the spontaneous recovery test. The model infers a new state is active upon extinction, due to the large prediction errors generated by the absence of the expected reward. The extinction state is believed to remain active, as the agent is biased to believing the most recently experienced latent state is active. The time delay has the effect of a context shift, such that the agent's beliefs about which latent state is active are reset. Therefore, the agent now believes that it is equally likely that the conditioning state is active relative to the extinction state. This is reflected in the brief increase in associative strength. Thus, the model can successfully capture the decrement in performance associated with non-reinforcement of a conditioned CS as well as the recovery of performance that comes with passage of time.

### Gradual extinction

*Appendix 3—figure 1C, D* display simulations in relation to a gradual extinction protocol. The conditioning and spontaneous recovery stages were identical to the standard extinction simulation. The extinction stage was split into four separate blocks (t=11–30; t=31–50; t=51–70; t=71–90). Across the first three blocks, the CS continued to be paired with reward; however, the reward value (intensity) was progressively decreased between blocks: it was halved across the first three blocks before being omitted altogether (as it was in the empirical experiments), such that R=0.5 for block one (t=11–30), R=0.25 for block two (t=31–50), R=0.125 for block three (t=51–70) and, finally, R=0

(reward removed) for block four (t=71–90). Associative strength steadily declined across extinction as the reward intensity decreased. Critically, at no point was a new state inferred as prediction error magnitude did not reach the threshold necessary for a new state. Therefore, the agent only has a single state in the spontaneous recovery stage and does not display any recovery of performance. Thus, the model can explain why gradual extinction results in robust extinction learning such that performance does not recover with the passage of time.

### Scrambled extinction

The model predicts that the gradual extinction effect depends on an ordered reduction in the reward intensity. Specifically, it predicts that presenting the same number of lower-intensity rewards in a pseudo-random order would not allow for conditioning and extinction to be encoded in the same state, which is the very basis of the gradual extinction effect according to latent state models. Simulations support this prediction. *Appendix 2—figure 1E, F* show simulations of a scrambled extinction protocol. The conditioning and spontaneous recovery stages were identical to those stages in the previous simulations. The extinction stage was once again split into four blocks as in the gradual extinction simulation. The first three blocks contained equal amounts of each lower-intensity reward used in the gradual extinction simulation (e.g. 7 × R=0.5, 7 × R=0.25, 6 × R=0.125 for block 1 [block 1 contained 6 × R=0.125, block 2 contained 6 × R=0.5, block 3 contained 6 × R=0.25]). Block 4 once again had the reward removed (R=0).

Associative strength decreased somewhat upon entering extinction and then, was roughly maintained as the reward value jumped around before finally decreasing to zero. Critically, the lack of ordered reduction created large predictions errors: hence, a new state was inferred to be active when the reward was fully removed (final block of extinction), and associative strength increased in the test stage for spontaneous recovery. Thus, the model shows that scrambled extinction, achieved by presenting the lower-intensity shocks in a pseudo-random order, is not sufficient to produce robust extinction learning.

### Weak shock extinction

The model further predicts that the gradual extinction effect is dependent on a gradual reduction in reward intensity, as opposed to an abrupt removal. Abruptly decreasing the reward intensity will cause state-splitting and will not allow for conditioning and extinction to be encoded in the same state. *Appendix 3—figure 2A, B* display simulations in relation to a weak shock extinction protocol. The conditioning and spontaneous recovery stages were identical to those stages in the previous simulations. The extinction stage was split into four blocks where the first three blocks had R=0.125 (t=11–70, R=0.125), the lowest value reward used in the gradual extinction simulation. The final block had the reward removed (t=71–90, R=0).

Associative strength decreased sharply upon entering extinction and further decreased to 0 once the reward was removed. The large prediction error created by the abrupt reduction in reward intensity caused a new state to be inferred, leading to a recovery of associative strength in the final spontaneous recovery test. Thus, the model explains why a gradual reduction in reward intensity is necessary to observe the gradual extinction effect, as an abrupt reduction in reward intensity produced a recovery of performance after a time delay.

## Contextual shifts and gradual extinction

The model conceptualizes context shifts (i.e. shifting the physical and/or temporal context) as a resetting of an agent's beliefs in latent states. That is, when the context is shifted, beliefs are reset to be uniform, such that all known latent states are equally likely to be active (i.e. the degree of belief in a single state is 1 /L [L=total number of latent states]). This removes the agent's bias to believing the state active on the most recent trial is likely to be active again. Critically, context shifts do not affect the state-splitting process. Only prediction errors relating to the CS-US relationship are able to determine whether a new state is inferred to be active or not. Thus, despite a context shift resetting current beliefs about active latent states, the model predicts that combining gradual extinction with a shift in context (physical or temporal shift) will not attenuate the gradual extinction effect. *Appendix 3—figure 2C, D* display simulations of gradual extinction with a physical context shift; and *Appendix 3—figure 2E, F* display simulations of gradual extinction with a temporal context shift.

A context shift is instantiated in the model as resetting beliefs to 1 /L for a physical context shift (L=number of latent states). For a temporal shift, as the chosen time interval grows, the beliefs

approach 1 /L. The chosen interval for the simulations was t=200, effectively fully resetting beliefs to 1 /L. The context shift (temporal and physical) occurred between conditioning and extinction, the simulations remained identical to the gradual extinction simulation otherwise.

The pattern of associative strength was identical to the gradual extinction simulation, as the context shift had no impact on whether a new state was inferred. Thus, as there is only a single state at the spontaneous recovery stage, associative strength did not recover. Thus, the model predicts that the robust learning produced by gradual extinction is not affected by a physical or temporal context shift. However, the results of Experiments 5 and 6 contradict this prediction as they show that a context shift (physical or temporal) attenuates the effectiveness of gradual extinction. Therefore, an expanded role for context would be a worthwhile addition to *Cochran and Cisler, 2019* latent state model.

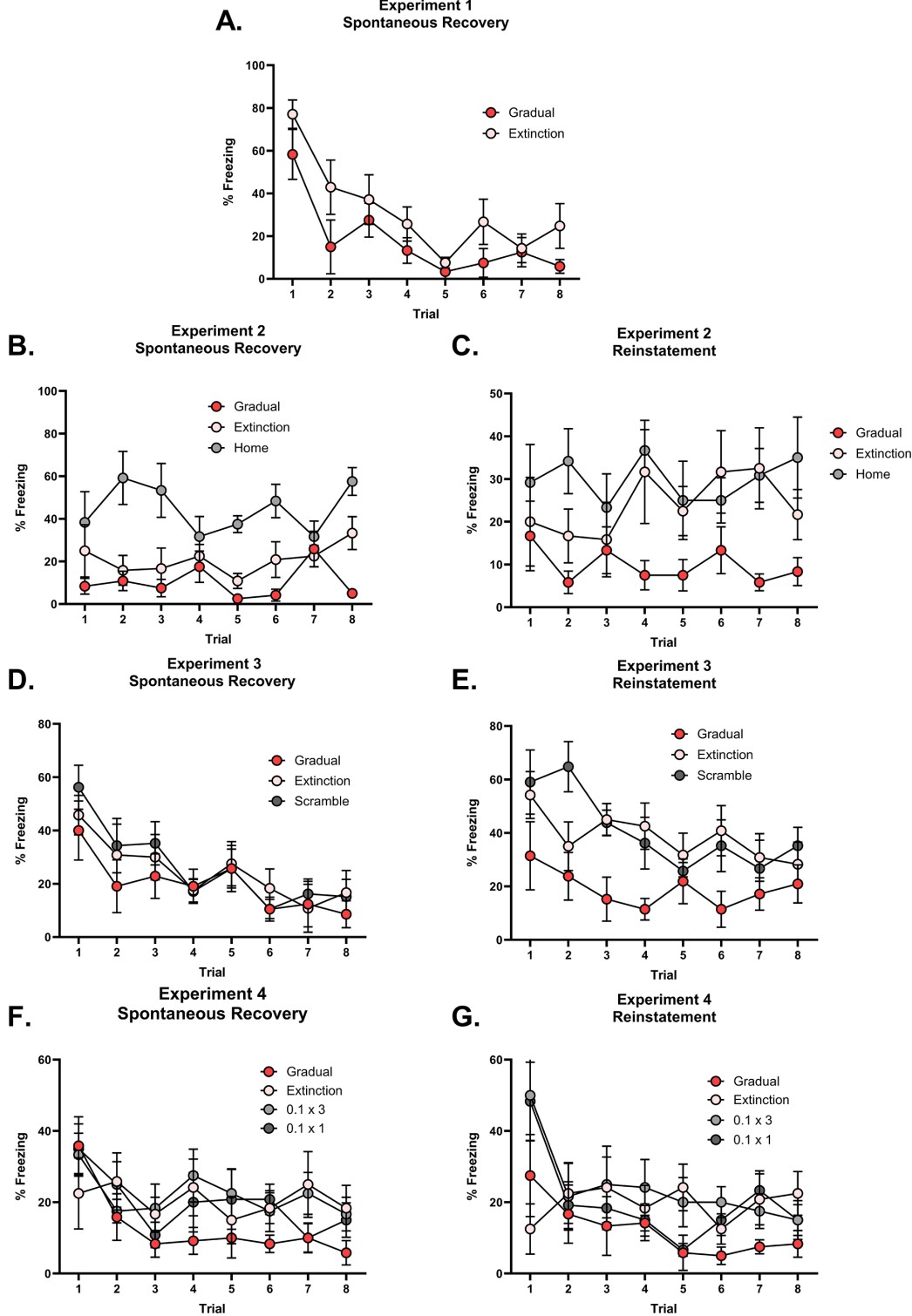

**Appendix 3—figure 1.** Simulations of the *Cochran and Cisler, 2019* latent state model for designs used in Experiments 4–6. (**A**). Associative strength (AS) simulation of a conditioned stimulus (CS) that receives extinction and testing for spontaneous recovery. Associative strength increases across conditioning, decreases across extinction before recovering at test, after a time delay, indicating spontaneous recovery. (**B**). Latent state beliefs (LSB) of a CS that receives extinction and testing for spontaneous recovery. Latent state beliefs switch at the beginning of extinction indicating that a new state has been inferred (due to the large prediction error caused by the absence of the reward). (**C**). Associative strength simulation of a CS that receives gradual extinction and testing

*Appendix 3—figure 1 continued on next page*

*Appendix 3—figure 1 continued*

for spontaneous recovery. Associative strength increases across conditioning, decreases steadily across gradual extinction, and remains low after a time delay, indicating an absence of spontaneous recovery. (**D**). Latent state beliefs of a CS that receives gradual extinction and testing for spontaneous recovery. Latent state beliefs do not change across the simulation, indicating that all stages were encoded into the same state. (**E**). Associative strength simulation of a CS that receives scrambled extinction and testing for spontaneous recovery. Associative strength increases across conditioning, decreases steadily across scrambled extinction (with high trial-to-trial variability) before recovering briefly after a time delay, indicating spontaneous recovery. (**F**). Latent state beliefs of a CS that receives scrambled extinction and testing for spontaneous recovery. Latent state beliefs change upon the full removal of the reward in extinction (i.e. after the scrambled stage) indicating state-splitting and the necessity of a progressive reduction in shock intensity to ensure conditioning and extinction are encoded together.

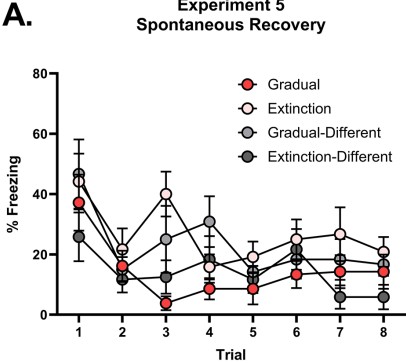

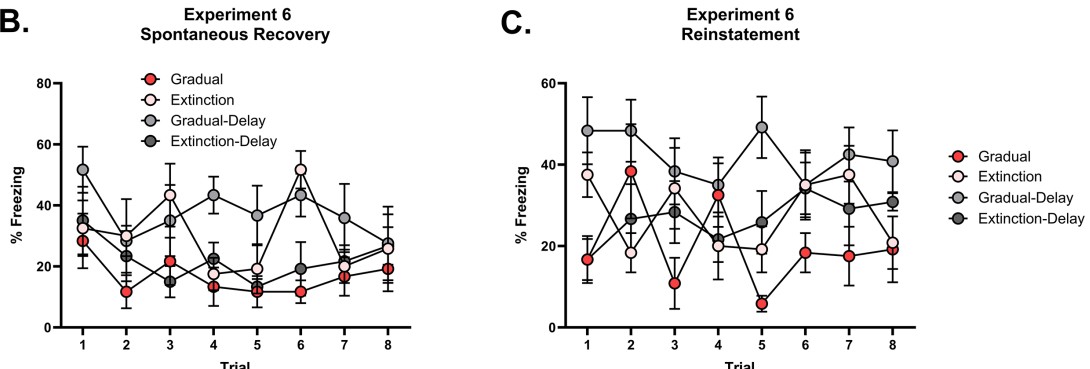

**Appendix 3—figure 2.** Simulations of the *Cochran and Cisler, 2019* latent state model for designs used in Experiments 4–6. (**A**) Associative strength (AS) simulation of a conditioned stimulus (CS) that receives weak shock extinction and testing for spontaneous recovery. Associative strength increases across conditioning, decreases quickly across presentations of the weak shock, and further decreases when the shock is removed before recovering at test, after a time delay, indicating spontaneous recovery. (**B**) Latent state beliefs (LSB) of a CS that receives weak shock extinction and testing for spontaneous recovery. Latent state beliefs switch at the beginning of weak shock extinction indicating that a new state has been inferred (due to the large prediction error caused by the decrease in shock intensity). (**C**) Associative strength simulation of a CS that receives gradual extinction in a different context from conditioning and testing for spontaneous recovery. Associative strength increases across conditioning, decreases steadily across gradual extinction, and remains low after a time delay, indicating an absence of spontaneous recovery. (**D**) Latent state beliefs of a CS that receives gradual extinction in a different context from conditioning and testing for spontaneous recovery. Latent state beliefs do not change across the simulation, indicating that all stages were encoded into the same state despite the physical context shift. (**E**) Associative strength simulation of a CS that receives gradual extinction after a time delay from conditioning and testing for spontaneous recovery. Associative strength increases across conditioning, decreases steadily across gradual extinction, and remains low after a time delay, indicating an absence of spontaneous recovery. (**F**) Latent state beliefs of a CS that receives gradual extinction after a time delay from conditioning and testing for spontaneous recovery. Latent state beliefs do not change across the simulation, indicating that all stages were encoded into the same state despite the temporal context shift.

**Appendix 3—table 1.** Adapted from *Cochran and Cisler, 2019*.
Parameters used for simulations of experiment designs 1–6.

| Parameter | Description | Value |
|---|---|---|
| $\alpha_0$ | Associative strength learning rate | 0.05 |
| $\beta_0$ | Variance learning rate | 0.05 |
| $\gamma_0$ | Latent-state transitions | 0.05 |
| $\sigma_0$ | Initial expected uncertainty | 0.5 |
| $\nu$ | Threshold for new state | 0.2 |
| $\delta$ | Unexpected uncertainty update | 0.6 |

