## [Editor Report · eLife assessment]

This is a **fundamental** study examining the role of prediction error in state allocation of memories. The data provided are **convincing** and largely support the conclusion that a gradual change between acquisition and extinction maintains the memory state of acquisition and thus results in extinction that is resistant to restoration. This paper is of interest to behavioural and neuroscience researchers studying learning, memory, and the neural mechanisms of those processes as well as to clinicians using extinction-based therapies in treating anxiety-based disorders

---

## [Referee Report · Reviewer #1 (Public Review)]

Summary:

In this study, Kennedy et al examine how new information is organized in memory. They tested an idea based on latent theory that suggests that large prediction error leads to the formation of a new memory, whereas small prediction error leads to memory updating. They directly tested the prediction by extinguishing fear conditioned rats with gradual extinction. For their experiment, gradual extinction was carried out by progressively reducing the intensity of shocks that were co-terminated with the CS, until the CS was presented alone. Doing so resulted in diminished spontaneous recovery and reinstatement compared to Standard Extinction. The results are compelling and have important implications for the field of fear learning and memory as well as translation to anxiety-related disorders.

The authors carried out the Spontaneous Recovery experiment in 2 separate experiments. In one, they found differences between the Gradual and Standard Extinction groups, but in the second, they did not. It seems that their reinstatement test was more robust, and showed significant differences between the Gradual and Standard Extinction groups.

The authors carried out important controls which enable proper contextualization of the findings. They included a "Home" group, in which rats received fear conditioning, but not an extinction manipulation. Relative to this group, the Gradual and Standard extinction groups showed a reduction in freezing.

In Experiments 3 and 4, the authors essentially carried out clever controls which served to examine whether shock devaluation (Experiment 4) and reduction in shock intensity (rather than a gradual decrease in shock intensity) (Experiment 3) would also yield a decrease in the return of fear. In-line with a latent-cause updating explanation for accounting for the Gradual Extinction, they did not.

In Experiment 5, the authors examined whether a prediction error produced by a change of context might contribute interference to the latent cause updating afforded by the Gradual Extinction. Such a prediction would align with a more flexible interpretation of a latent-cause model, such as those proposed by Redish (2007) and Gershman et al (2017), but not the latent-cause interpretation put forth by the Cochran-Cisler model (2019). Their findings showed that whereas Gradual Extinction carried out in the same context as acquisition resulted in less return of fear than Standard Extinction, it actually yielded a greater degree of return of fear when carried out in a different context, in support of the Redish and Gershman accounts, but not Cochran-Cisler.

Experiment 6 extended the findings from Experiment 5 in a different state-splitting modality: timing. In this experiment, the authors tested whether a shift in temporal context also influenced the gradual extinction effect. They thus carried out the extinction sessions 21 days after conditioning. They found that while Gradual Extinction was indeed effective when carried out one day after fear conditioning, it did not when conducted 21 days later.

The authors next carried out an omnibus analysis which included all the data from their 6 experiments, and found that overall, Gradual Extinction resulted in diminished return of fear relative to Standard Extinction. I thought the omnibus analysis was a great idea, and an appropriate way to do their data justice.

Strengths: Compelling findings. The data support the conclusions. 6 rigorous experiments were conducted which included clever controls. Data include male and female rats. I really liked the omnibus analysis.

Weaknesses: None noted

---

## [Referee Report · Reviewer #2 (Public Review)]

Summary:

The present article describes a series of experiments examining how a gradual reduction in unconditional stimulus intensity facilitates fear reduction and reduces relapse (spontaneous recovery and reinstatement) relative to a standard extinction procedure. The experiments provide compelling, if somewhat inconsistent, evidence of this effect and couch the results in a scholarly discussion surrounding how mechanisms of prediction error contribute to this effect.

Strengths:

The experiments are theoretically motivated and hypothesis-driven, well-designed, and appropriately conducted and analyzed. The results are clear and appropriately contextualized into the broader relevant literature. Further, the results are compelling and ask fundamental questions regarding how to persistently weaken fear behavior, which has both strong theoretical and real-world implications. I found the 'scrambled' experiment especially important in determining the mechanism through which this reduction in shock intensity persistently weakens fear behavior.

Weaknesses:

Overall, I found very few weaknesses with this paper. I think some might view the somewhat inconsistent effects on relapse between experiments to be a substantial weakness, I appreciate the authors directly confronting this and using it as an opportunity to aggregate data to look at general trends. Further, while Experiment 1 only used males, this was corrected in the rest of the experiments and therefore is not a substantial concern.

---

## [Referee Report · Reviewer #3 (Public Review)]

Summary:

The manuscript examined the role or large versus small prediction errors (PEs) in creating a state-based memory distinction between acquisition and extinction. The premise of the paper is based on theoretical claims and empirical findings that gradual changes between acquisition and extinction would lead to the potential overwriting of the acquisition memory with extinction, resulting in a more durable reduction in conditioned responding (i.e. more durable extinction effect). The paper tests the hypotheses in a series of elegant experiments in which the shock intensity is decreased across extinction sessions before non-reinforced CS presentations are given. Additional manipulations include context change, shock devaluation, controlling for lower shock intensity exposure. The critical comparison was standard non-reinforced extinction training. The critical tests were done in spontaneous recovery and reinstatement.

Strengths:

The findings are of tremendous importance in understanding how memories can be updated and reveal a well-defined role of PE in this process. It is well-established that PE is critical for learning, so delineating how PE is critical for generating memory states and the role it serves in keeping memories dissociable (or not) is exciting and clever. As such the paper addresses a fundamental question in the field.

The studies test clear and defined predictions derived from simulations of the state-belief model of Cochran & Cisler (2019). The designs are excellent: well-controlled and address the question.

The authors have done an excellent job at explaining the value of the latent state models.

The authors have studied both sexes in the studied presented, providing generality across the sexes in their findings. The figures depict the individual data points for males and females allowing the reader to see the responses for each sex.

The authors have addressed the previously raised weaknesses. They noted that procedurally it would be difficult to provide independent evidence that delivering a lower intensity shock will generate a smaller PE than say no shock. The differences in the data obtained based on error vs shock devaluation are convincing, although direct evidence for shock devaluation would have strengthened the argument.

---

## [Author Response]

The following is the authors’ response to the original reviews.

**Public Reviews:**

**Reviewer #1 (Public Review):**
Summary:In "Prediction error determines how memories are organized in the brain: a study of Pavlovian fear 2 extinction in rats", Kennedy et al examine how new information is organized in memory. They tested an idea based on latent theory that suggests that a large prediction error leads to the formation of a new memory, whereas a small prediction error leads to memory updating. They directly tested the prediction by extinguishing fear-conditioned rats with gradual extinction. For their experiment, gradual extinction was carried out by progressively reducing the intensity of shocks that were co-terminated with the CS, until the CS was presented alone. Doing so resulted in diminished spontaneous recovery and reinstatement compared to Standard Extinction. The results are compelling, and have important implications for the field of fear learning and memory as well as translation to anxiety-related disorders.The authors carried out the Spontaneous Recovery experiment in 2 separate experiments. In one, they found differences between the Gradual and Standard Extinction groups, but in the second, they did not. It seems that their reinstatement test was more robust, and showed significant differences between the Gradual and Standard Extinction groups.The authors carried out important controls that enable proper contextualization of the findings. They included a "Home" group, in which rats received fear conditioning, but not extinction manipulation. Relative to this group, the Gradual and Standard extinction groups showed a reduction in freezing.In Experiments 3 and 4, the authors essentially carried out clever controls that served to examine whether shock devaluation (Experiment 4) and reduction in shock intensity (rather than a gradual decrease in shock intensity) (Experiment 3) would also yield a decrease in the return of fear. In line with a latent-cause updating explanation for accounting for the Gradual Extinction, they did not.In Experiment 5, the authors examined whether a prediction error produced by a change of context might contribute interference to the latent cause updating afforded by the Gradual Extinction. Such a prediction would align with a more flexible interpretation of a latent-cause model, such as those proposed by Redish (2007) and Gershman et al (2017), but not the latent-cause interpretation put forth by the Cochran-Cisler model (2019). Their findings showed that whereas Gradual Extinction carried out in the same context as acquisition resulted in less return of fear than Standard Extinction, it actually yielded a greater degree of return of fear when carried out in a different context, in support of the Redish and Gershman accounts, but not Cochran-Cisler.Experiment 6 extended the findings from Experiment 5 in a different state-splitting modality: timing. In this experiment, the authors tested whether a shift in temporal context also influenced the gradual extinction effect. They thus carried out the extinction sessions 21 days after conditioning. They found that while Gradual Extinction was indeed effective when carried out one day after fear conditioning, it did not when conducted 21 days later.The authors next carried out an omnibus analysis which included all the data from their 6 experiments, and found that overall, Gradual Extinction resulted in diminished return of fear relative to Standard Extinction. I thought the omnibus analysis was a great idea and an appropriate way to do their data justice.Strengths:Compelling findings. The data support the conclusions. 6 rigorous experiments were conducted which included clever controls. Data include male and female rats. I really liked the omnibus analysis.

We thank the reviewer for their positive comments – they are appreciated.

Weaknesses:None noted
**Reviewer #2 (Public Review):**
Summary:The present article describes a series of experiments examining how a gradual reduction in unconditional stimulus intensity facilitates fear reduction and reduces relapse (spontaneous recovery and reinstatement) relative to a standard extinction procedure. The experiments provide compelling, if somewhat inconsistent, evidence of this effect and couch the results in a scholarly discussion surrounding how mechanisms of prediction error contribute to this effect.Strengths:The experiments are theoretically motivated and hypothesis-driven, well-designed, and appropriately conducted and analyzed. The results are clear and appropriately contextualized into the broader relevant literature. Further, the results are compelling and ask fundamental questions regarding how to persistently weaken fear behavior, which has both strong theoretical and real-world implications. I found the 'scrambled' experiment especially important in determining the mechanism through which this reduction in shock intensity persistently weakens fear behavior.

We thank the reviewer for their positive comments – they are appreciated.

Weaknesses:Overall, I found very few weaknesses in this paper. I think some might view the somewhat inconsistent effects on relapse between experiments to be a substantial weakness, I appreciate the authors directly confronting this and using it as an opportunity to aggregate data to look at general trends. Further, while Experiment 1 only used males, this was corrected in the rest of the experiments and therefore is not a substantial concern.
**Reviewer #3 (Public Review):**
Summary:The manuscript examined the role of large versus small prediction errors (PEs) in creating a state-based memory distinction between acquisition and extinction. The premise of the paper is based on theoretical claims and empirical findings that gradual changes between acquisition and extinction would lead to the potential overwriting of the acquisition memory with extinction, resulting in a more durable reduction in conditioned responding (i.e. more durable extinction effect). The paper tests the hypotheses in a series of elegant experiments in which the shock intensity is decreased across extinction sessions before non-reinforced CS presentations are given. Additional manipulations include context change, shock devaluation, and controlling for lower shock intensity exposure. The critical comparison was standard non-reinforced extinction training. The critical tests were done in spontaneous recovery and reinstatement.Strengths:The findings are of tremendous importance in understanding how memories can be updated and reveal a well-defined role of PE in this process. It is well-established that PE is critical for learning, so delineating how PE is critical for generating memory states and the role it serves in keeping memories dissociable (or not) is exciting and clever. As such the paper addresses a fundamental question in the field.The studies test clear and defined predictions derived from simulations of the state-belief model of Cochran & Cisler (2019). The designs are excellent: well-controlled and address the question.The authors have done an excellent job of explaining the value of the latent state models.The authors have studied both sexes in the study presented, providing generality across the sexes in their findings. However, depicting the individual data points in the bar graphs and noting which data represent males and which represent females would be of great value.

We thank the reviewer for their positive comments. We have included individual data points in the bar graphs and indicated which represent males and females.

Weaknesses:(1) While it seems obvious that delivering a lower intensity shock will generate a smaller PE than say no shock, it would have been nice to see data from say a compound testing procedure that confirms this.

It would be great if we could provide independent evidence that shifting from a 0.8 mA shock to a 0.4 mA shock (first session of gradual extinction) produces a smaller prediction error than shifting from a 0.8 mA shock to no shock at all (first session of standard extinction). In theory, this could be assessed using Rescorla’s (2000) compound test procedure. However, application of this procedure requires the use of a within-subject design and latent state theories would not predict the gradual extinction effect in such a design (as all prediction errors generated in such a design would affect the state-splitting process). That is, the between-subject design used to generate the gradual extinction effect is not amenable to application of the compound test procedure; and the within-subject design in which the compound test procedure could be applied is unlikely to generate the gradual extinction effect. Thus, we instead rely on the high degree of similarity between our results and those predicted by Cochran & Cisler (2019) to argue that the gradual extinction protocol produces a series of smaller prediction errors than does the standard extinction protocol: hence the present pattern of results.

(2) The devaluation experiment is quite clever, but it also would be strengthened if there was evidence in the paper that this procedure does indeed lead to shock devaluation.

The aim of Experiment 3 was to determine whether the gradual extinction effect is due to prediction error-based memory updating or shock devaluation. If the effect was due to shock devaluation, the group that received the gradual extinction treatment should have displayed the same low level of spontaneous recovery as the group that only experienced the shock at its lowest (0.1 mA) intensity (i.e., the shock devaluation group). Contrary to this prediction, the results showed that the gradually extinguished group displayed *less* spontaneous recovery than the shock devaluation group. That is, in this experiment, the slow and progressive reduction in shock intensity *was* processed differently to the repeated 0.1 mA shock exposures but the results were inconsistent with any shock devaluation effect. Hence, we conclude that the gradual extinction effect does not involve shock devaluation but instead is due to prediction error-based memory updating.

(3) It would have been very exciting to see even more parametric examinations of this idea, like maintaining shock intensity but gradually reducing shock duration, which would have increased the impact of the paper.

We appreciate the reviewer’s point. As each shock was presented for just 0.5 s, we are not confident that rats would detect gradual and progressive changes in its duration in the same way as they can obviously detect gradual and progressive changes in its intensity. We are, however, investigating the effects of gradual extinction in a second order conditioning protocol, which will allow us to examine the full range of parameters that are important for its regulation, including manipulations of stimulus duration. In our second-order conditioning protocol, rats are first exposed to pairings of a 10 s S1 and a 0.5 s foot shock US; and then exposed to pairings of a 30 s S2 and the 10 s S1. Across the latter pairings, rats acquire second-order conditioned fear responses to S2. Importantly, these responses can be extinguished through repeated presentations of the S2 in the absence of its S1-associate; and the duration of the S1 can be progressively and gradually reduced from 10 s to 0 s across the shift to this extinction. These experiments are currently in progress and will eventually represent an extension of the present findings.

(4) Individual data points should be represented in the test figures (see above also).

We have updated the figures to show these data points.

Rescorla, R. A. (2000). Associative changes in excitors and inhibitors differ when they are conditioned in compound. *Journal of Experimental Psychology: Animal Behavior Processes*, *26*(4), 428.

**Reviewing Editor (Recommendations For The Authors):**
The eLife assessment relates to the present form of the paper. However, following a discussion with the reviewers, the significance of the findings could be bolstered to fundamental if you decided to revise the current manuscript by scaling up the investigation to examine a wider set of parameters and conditions under which error can influence state allocation of memories. One way of doing this, but not limited to this, is suggested in the reviews (e.g. maintaining shock intensity, reducing its duration). Relatedly, a more extensive discussion of the Gershamn et al. (2013) paper would be relevant.

As noted in our response to Reviewer 3, we are currently investigating the effects of gradual extinction in a second order conditioning protocol, which will allow us to examine the full range of parameters that are important for its regulation, including manipulations of stimulus duration. These experiments are in-progress and will eventually represent an extension of the present findings. They are not, however, ready to be included as part of the present study.

We have further referenced the Gershman et al., (2013) paper as well as the related Bouton et al., (2004) paper on the effects of gradually reducing the frequency of the US across extinction. This appears in the fifth paragraph of the Discussion: “The present study adds to a growing body of evidence that manipulations applied across the shift from CS-US pairings to presentations of the CS alone can influence the effectiveness of extinction. For example, Gershman et al., (2013) and Bouton et al., (2004) showed that gradually reducing the proportion of reinforced CS presentations results in less spontaneous recovery and slower reacquisition, respectively; though both studies left open fundamental questions about the basis of their findings (see also Woods & Bouton, 2007).”

**Reviewer #1 (Recommendations For The Authors):**
I don't have any strong recommendations. I think the paper is really great as is.One minor suggestion to consider:The authors carried out the Spontaneous Recovery experiment in 2 separate experiments. In one, they found differences between the Gradual and Standard Extinction groups, but in the second, they did not. This is perhaps not entirely surprising, since their extinction test was conducted 2 weeks post-extinction, and not all rats show spontaneous recovery within that timeframe. The authors mention that the lack of SR might be due to the low level of freezing reported in their test, but since they are showing group mean data, they might consider showing the individual data points to showcase the range of SR freezing as an additional way to make sense of the variability (ie, maybe a few rats that showed very low freezing carried the mean down in the Standard Extinction group, while others showed return of fear).

We agree and have included individual data points for test results in Figures 2D, 2F, 3D, 3H, 4D and 4H. Hence, these figures now reflect both group and individual freezing levels.

**Reviewer #2 (Recommendations For The Authors):**
Overall, I thought this was an exceptional paper. Aside from the comments listed above which I'm not sure are inherently addressable, the only real changes I would like to see are that individual data points should be depicted in the main testing figures, as is becoming more conventional in the field.

We thank the reviewer for their positive comments. As indicated in our response to the other reviewers, we have added individual data points to the histograms showing test results.

**Reviewer #3 (Recommendations For The Authors):**
Figures(1) The test data are presented as bars, but I did wonder if there were differences between the groups from the start of testing or if those emerged across testing (SR vs extinction savings).

We have added two new figures to the supplementary section, Figures 8 and 9. These display the trial-by-trial data from spontaneous recovery and reinstatements tests in each experiment. The data clearly show that the between-group differences in freezing were very stable across the test sessions.

(2) While I understand the importance of presenting the last extinction session, I felt depicting the entire CS session would be more informative. Alternatively, removing this altogether and leaving the information from the extinction session in the supplemental would focus the reader on the key test data.

We appreciate the reviewer’s point. It is important to show that the groups displayed equivalent freezing in the final extinction session prior to testing. Given that the test data are conveniently and best presented in a histogram, we have chosen to present the data from the final extinction session in the same way. The full, trial-by-trial trajectory of freezing across conditioning and extinction, as well as the analyses of these data, are presented in the supplementary A.

(3) I did not find the figures to be very aesthetically pleasing (in part because some panels were unnecessarily large). For example, I found it rather odd that the simulation panels were split in Figure 1. One suggestion of how this figure could look better is to keep the size of panels B, C, and D the same and align them on the same row with the design figure above them. The other option is to have the design figure above the test figure and the two simulation figures above each other and next to the design and test. Also, there are grey lines that appear around the simulation figures on my PDF.

We have updated the figures so that they are consistent across experiments and more aesthetically pleasing. Specifically, we have consistently: (1) inserted the simulations of Cochran & Cisler (2019) next to the design schematic; (2) inserted the extinction and test data beneath the design schematic; and (3) Made the sizing of figures more uniform across Experiments 1-6.